

# On the proper use of temperature screen-level measurements in weather forecasting models over mountains

Danaé Préaux[1,*], Ingrid Dombrowski-Etchevers[1,*], Isabelle Gouttevin[2,*], and Yann Seity[3]

[1]CNRM, Université de Toulouse, Météo-France, CNRS, Toulouse, France
[2]Univ. Grenoble Alpes, Université de Toulouse, Météo-France, CNRS, CNRM, Centre d'Etudes de la Neige, Grenoble, France
[3]Météo-France, DIRSO/CMP, Foix, France
[*]These authors contributed equally to this work.

**Correspondence:** Danaé Préaux (danae.preaux@meteo.fr)

**Abstract.** Near surface air temperature, considered to be 2m above the ground, is a key meteorological parameter with a wealth of uses for mankind. However, its accurate estimation in mountain regions is impeded by persistent limits inherent to atmospheric modelling over complex terrain. In the present study, we analyze the role of structural inhomogeneities of the valleys and mountains observational network in France, to highlight their contribution to the misrepresentation of near-surface

air temperature over mountain regions in the numerical weather prediction (NWP) system Arome-France. We scrutinized the disparity in height above ground of the temperature measurements, the inhomogeneous geographical distribution of stations that are preferentially located in valleys, and the relief mismatch between station location and model grid points. The consequences of these inhomogeneities are analyzed for model evaluation and throughout the data assimilation process. In France, high altitude stations usually measure temperature at about 7m over the snow-free ground, and on average one meter lower when the

ground is snow-covered in winter. We show this height difference with respect to standard stations measuring at 2m, should be considered when evaluating the model performances and in assimilation. We show that due to the current 3DVar assimilation system, the assimilation of valley stations affects the near-surface temperature analysis at all altitudes in the mountains. The altitude mismatch between observation points and model grid points does not play an important rôle, probably in part due to its relatively marginal occurrence in an NWP system with 1.3 km grid spacing. In summary, this study describes new methods

for comparing models with mountain observation data, both in terms of assimilation and performance assessment.

## 1  Introduction

In mountain regions, the knowledge and forecast of near-surface air temperature is key to numerous socio-economic applications ranging from natural hazards (Morin et al., 2020; Vionnet et al., 2020) to recreational activities Becken (2010) and agriculture and water resource management (Spandre et al., 2016; Jörg-Hess et al., 2015). In this latter respect, near-surface

air temperature, sometimes referred to as screen-level temperature, is often used in hydrological models for the partitioning of precipitation between rain and snow. Temperature is furthermore one of the key variables of sensible weather, contributing



to shape ecosystems and human implantation. It is a primary essential climate variable for climate monitoring and assessment (IPCC 2021) and its accurate description over mountain regions is a prerequisite for any climatic study in these environments.

Atmospheric models, in particular high resolution numerical (NWP) weather prediction models, are routinely used by me-
teorological centers to simulate and forecast spatially distributed screen-level temperatures at regional scales. These models often share important parts of their structures, characteristics and behaviors with regional climate models (RCM) (Pichelli et al., 2021; Torma et al., 2015). However, both types of models exhibit significant biases over mountain regions, limiting their relevance for a variety of uses they are originally designed for Rudisill et al. (2024); Gouttevin et al. (2023). For instance, Monteiro et al. (2022) identified a spurious snow accumulation bias in their climatic simulations performed with the CNRM-
Arome RCM, that preclude any analysis of the results above 2500m altitude in the French Alps. The authors analyze that this bias could proceed from several origins, among which a pronounced cold bias for mountain regions, affecting both the NWP (Arome-France) and RCM (CNRM-Arome) versions of Arome (Application de la Recherche Opérationnelle à Méso-Échelle). In their extensive review study of the temperature biases of regional-scale, high resolution atmospheric models over mountain regions, Rudisill et al. (2024) highlight that a cold near surface bias over altitude regions is the most general behaviour for
such models over all mid-latitude, snow-covered mountain regions of the world. While primarily strong over peaks and ridges, it often comes with a warm bias in valleys. These characteristics are precisely the ones observed for the Arome-France high resolution NWP system (hereafter just 'Arome') used for operational weather forecasting in France. A literature review complemented by the scrutiny of forecasters' reports in mountain regions, enables a more precise decomposition into three types of biases: (1) a cold bias at high altitude, (2) a low-altitude warm bias occurring in stably stratified layers and (3) a warm bias
during snowfall situations.

The warm bias in valleys appears during long-lasting anticyclonic situations in winter. It was highlighted during the 2015 observational campaign held in Passy, in the Arve Valley (Northern French Alps, Paci et al., 2016)). This campaign revealed that the warm biases of the model during such situations, hinder the forecasting and representation of the pollution events often affecting alpine valleys in winter due to strong traffic, wood fire heating Aymoz et al. (2007), and poor air mixing. The
air temperature is a key meteorological parameter for the construction of winter pollution risk indicators Paci et al. (2016), enhancing the need for its accurate estimation. The second Arome warm bias manifests in valleys when a warm front meets the relief, especially in the direction perpendicular to the valley. In these situations the modelled rise in temperature is too strong with respect to what is observed. This is often accompanied by an altitudinal elevation of the snowfall line in the model, that can leads to misidentifying the precipitation type (rain in the model instead of snow) or underestimating the quantity of snow
precipitation in valleys, where the major roads are. This issue is not new. In its internal report on the Arome model behavior over the winter 2017-2018, Beauvais (2018) describes three of such events with a snowfall/rainfall partition problem while mentioning similar situations dating back to 2009.

Finally, a cold bias increasing with altitude was originally detected by Vionnet et al. (2016) in a previous version of Arome, that ran at 2.5 km over France with 60 vertical levels. Temperature data collected over 4 years (2010-2014) at 33 stations in
the French Alps, revealed differences to the model of -0.5 °C below 1500m, but that reached 3 °C at night between 1500 and 2500m altitude. Above this altitude, the mean bias is over 3 °C in winter at night, and just less than 2 °C during daytime. This



bias exhibits a strong seasonality, being more important in winter, when the snow cover dominates at high altitude, than in summer (Dombrowski-Etchevers et al., 2017). This bias was confirmed by Gouttevin et al. (2023) in the current operational Arome model version running at 1.3 km with 90 vertical levels. The bias has strong implications for the modeled snowpack,

in particular leading to too high snow accumulations (mentioned above) and also delayed snow melt disqualifying its use in support of water resource management and possibly, flood forecasting. This also prevents the use of the model to provide the atmospheric conditions to avalanche-warning dedicated snow models, as the snowpack evolution and the formation of weak layers often involved in avalanche activity, is particularly sensitive to vertical thermal gradients (Gouttevin et al., 2018).

As described in the studies cited above, in-situ observations are often used to evaluate models and provide bias assessments

or skill scores that routinely accompany the development of NWP model. In this process, the change of a parametrization, a modification in the dynamics or general model setup, is only accepted if it doesn't degrade operational scores. However, a feature poorly considered by model developers in this process, are the specificities inherent to mountain environment, that have key implications on the measurements carried out there and on their suitability for use in routine model evaluation without any adaptation. One of such specificities in mid-latitude regions is snow. Due to the development of a quite thick snowpack

in mid-latitude, alpine regions (e.g. Sturm and Liston, 2021), temperature measurements are generally not at a constant height above the (possibly snow-covered) surface. Nor is it between 1.25 and 2m height above ground as recommended by the WMO (a standard often ignored by modelers who generally consider the measurement to be at 2m). To limit the risk that sensors get covered in snow during winter, screen-level temperature observations are usually made at a higher height above the snow-free ground in altitude regions than in valley/plain environments. This is typically the case in France, where the sensors of

the Nivose stations designed for the high-altitude mountain, are about 7.5m above the snow-free ground. This is also the case in e.g. Switzerland where the IMIS stations (Intercantonal Measurement and Information System) used among others by Meteo-Swiss, can be as high as 6m above snow-free ground (https://www.slf.ch/en/avalanche-bulletin-and-snow-situation/measured-values/description-of-automated-stations/). However, to the best of our knowledge, this height difference is not accounted for when either operational scores (at least at the French Met Service) or academic model evaluations are performed.

While examining the majority of the references cited by (Rudisill et al., 2024), we could not find any mention of observation vs model height adjustment for temperature comparisons, even in seasonally snow covered regions. Required adjustments for altitudinal mismatch between model grid and station location, are much more commonly found in the literature and have been a preoccupation for numerous modelers (e.g. Rudisill et al., 2024; Quéno et al., 2016), kind of covering up the height-above-surface adjustments that we here mention. The true height above snow-free ground of the Nivose sensors, was not accounted

for in Vionnet et al. (2016) nor Dombrowski-Etchevers et al. (2017). As an answer to this knowledge gap, the first result section of this manuscript will focus on the lower boundary layer thermal dynamics at a mid-altitude and high-altitude instrumented sites and examine the differences between 2m temperature measurements and observations carried out higher up, deciphering the implications of the height-above-surface mismatch for model evaluation.

Another fundamental use of screen-level temperature observations in operational NWP, is their assimilation to enhance

the representation of the atmospheric state prior to forecasting its evolution over the upcoming hours and sometimes days (Brousseau et al., 2016; Demortier et al., 2023; Guillet, 2019; Gustafsson et al., 2018). As a matter of fact, the progress of



NWP systems in recent decades has been much driven by the increase in data assimilation, especially relying on satellite data (e.g. Fischer et al., 2018). In Arome-France, screen-level temperature observations are used in two different assimilation systems (Fig 1 and section 2.1), respectively majorly affecting the surface (Marimbordes et al., 2024) and the atmosphere

(Brousseau et al., 2016). However, the height-above-ground specificies of high-altitude stations, are not accounted for in either assimilation systems. Therefore, one aspect of the present study is also dedicated to estimate the impacts of this specificity in the assimilation, focusing on the atmospheric part and in link with the near-surface temperature biases highlighted earlier.

Finally, in-situ observations from mountain regions are inherently heterogeneous when it comes to their topographic context. Most of them are in valleys or mid-altitude mountain, where accessibility and maintenance are made easier (e.g. Vernay et al.,

2022; Thornton et al., 2022). While model evaluations in complex terrain regions quite often discriminate results either into altitude bands (e.g. Vionnet et al., 2016; Monteiro et al., 2022) or classes derived from landforms (ridges, crests, valleys, plains..., e.g. Winstral et al., 2017), such distinctions are not made in the assimilation. The structure functions that propagate the analysis increment spatially, do sometimes account for the topographic and landform heterogeneities (e.g. Deng and Stull, 2005), but this is not the case in Arome for the 3D-var atmospheric assimilation system. In the present study, we also scrutinize

how the spatial heterogeneity of the observation network with respect to topography affect the quality and efficiency of screen-level temperature observations into the Arome NWP system.

In a nutshell, the present study draws the light on some pitfalls affecting the use of the near-surface air temperature observations in mountain terrain for numerical weather forecasting. Taking the example of the Arome-France NWP system that operationally runs over a large alpine region, we aim at quantifying the impact on the model performance assessment and in

assimilation, of varied sensors'height above surface, relief mismatch between observations and model, and valley-vs-mountain heterogeneities in observational density. The plan of our manuscript addresses these items sequentially, after a section dedicated to material, method and study area. To the best of our knowledge these questions have not thoroughly been addressed in mid-latitude mountain regions of the world. We also take the opportunity to propose perspectives that circumvent the problems highlighted, for the benefice of weather forecasting in complex terrain.

## 2 Material and Methods

### 2.1 The Arome numerical weather prediction system and its assimilation

The limited-area NWP model Arome has been operational since December 2008 and runs over the domain named "France", illustrated in Figure 2. It is coupled to the French global model Arpege (Action de Recherche Petite Échelle Grande Échelle) which has a variable spectral mesh (Courtier and Geleyn, 1988) and improved resolution over Europe. Initially with a horizontal

resolution of 2.5 km (Seity et al., 2011), Arome has been producing forecasts on a 1.3 km grid since April 2015 (Brousseau et al., 2016). Its physics is the same as Meso-NH (Mesoscale NonHydrostatic Model) (Lafore et al., 1998; Lac et al., 2018). Thus it is a non-hydrostatic model, i.e. it "explicitly solves the system of compressible Euler equations without neglecting the vertical acceleration in the continuity equation, which allows a better representation of vertical motions or orography". (extract from Arnould et al., 2021). Arome uses the dynamics of the Aladin (Adaptation Dynamique Développement International)





model (Bubnová et al., 1995). Although the first version of Arome had 60 vertical levels with the first level at 10m above the surface, the version now used operationally has 90 vertical, the first of which is between 4.5m and 5.5m in the model, depending on weather conditions, i.e. approximately 5m. As a research option, a version of this model is available with 500m horizontal resolution and/or 120 or 156 vertical levels (with lowest level at 2.5m approximately).

For the surface scheme, Arome is coupled to Surfex (Surface EXternalised) (orange boxes in Figure 1) (Masson et al., 2013),

with, for vegetation, the Isba (Interaction Soil-Biosphere-Atmosphere) scheme (Noilhan and Planton, 1989) and, for snow, the D95 single-layer scheme (Douville et al., 1995).

To ensure that the model is as close as possible to the real state of the atmosphere, it is regularly corrected using observations. This process is called data assimilation and is described for near-surface temperatures in Figure 1. For simplicity in the following, we will refer to near-surface air temperature as T2m, despite the fact that it is conventionally measured between

1.25 and 2m above the surface following the WMO standards, and between 1.5m and 2m according to the French Met service standards. When refering to modeled values for near-surface temperatures, we will also use the term T2m (often with the suffix "_mod"). In that case T2m refers to a temperature diagnostic produced by the model for a 2m height above the surface.

In Arome, the assimilation takes place both in the atmosphere and in the surface (the blue and green boxes in Figure 1), respectively), but without interaction. In addition, the assimilation methods differ. Furthermore, the presence of fields dating

from before the use of Surfex is necessary for the assimilation to run smoothly, whether for the atmosphere or the surface (greyed-out box in Figure 1).

For convenience, in the diagram and in the rest of the article, T5m_ mod refers abusively to the temperature at the first level of the model. The surface temperature (Ts_ mod) corresponds to the surface temperature of the ground for Arome. If this ground is snow-covered, then it becomes the surface temperature of the snow cover (Giard and Bazile, 2000). T5m_mod

and Ts_mod are prognostic variables. These two temperatures are used to compute T2m_mod according to Geleyn (1988)'s diagnostic.

### 2.1.1   The 3DVar altitude assimilation

The assimilation of atmospheric variables in Arome is based on the 3DVAR (Three-dimensional Variational system) (red box of Figure 1), with an hourly data assimilation cycle (Brousseau et al., 2016; Gustafsson et al., 2018). The aim is to minimize the

difference between the observations and the guess (Guillet, 2019), which in this case corresponds to a 1 h Arome forecast (box entitled "T2m (diag)(P1)" of Figure 1) calculated before each analysis on the basis of the previous analysis (purple-bordered box entitled "T2m (diag)(P0)" of Figure 1). To do this, all observations, whether satellite or surface, are first subjected to a quality control known as screening. This stage eliminates observations that are considered doubtful because they come from a non-qualified source or are too far away from the design. However, if this guess is biased, screening can also unfairly reject

certain observations.



After screening, the 3DVar combines the observations with the guess (orange-bordered box entitled "T2m (guess at obs point)" in Figure 1) to produce the new analysis by minimizing the cost function J (Demortier et al., 2023):

$$J(x) = \frac{1}{2}(x - x_b)^{\mathrm{T}}\mathbf{B}^{-1}(x - x_b) + \frac{1}{2}[y_o + \mathcal{H}x]^{\mathrm{T}}\mathbf{R}^{-1}[y_o + \mathcal{H}x],$$

where $x_b$ corresponds to the background state, $y_o$ to the observation vector, $\mathcal{H}$ to the (non-linear) observation operator which allows different types of information to be compared, $\mathbf{R}$ and $\mathbf{B}$ are the observation and background error covariance matrices and the exponent T to the transpose of the associated matrix. The matrix $\mathbf{B}$ contains background error covariances in the spectral space. This dependence on the spatial neighborhood depends on the correlation lengths of the errors, which in Arome

are spatially uniform and do not take into account relief. Furthermore, this $\mathbf{B}$ matrix is constant in time.

Increments are calculated for the surface observations and for the upper-air observations. Then, $J$ is minimized using these increments. However, the increment of in-situ observations is calculated at 2m, but is not carried upstream to the height of the first level of the model before being used.

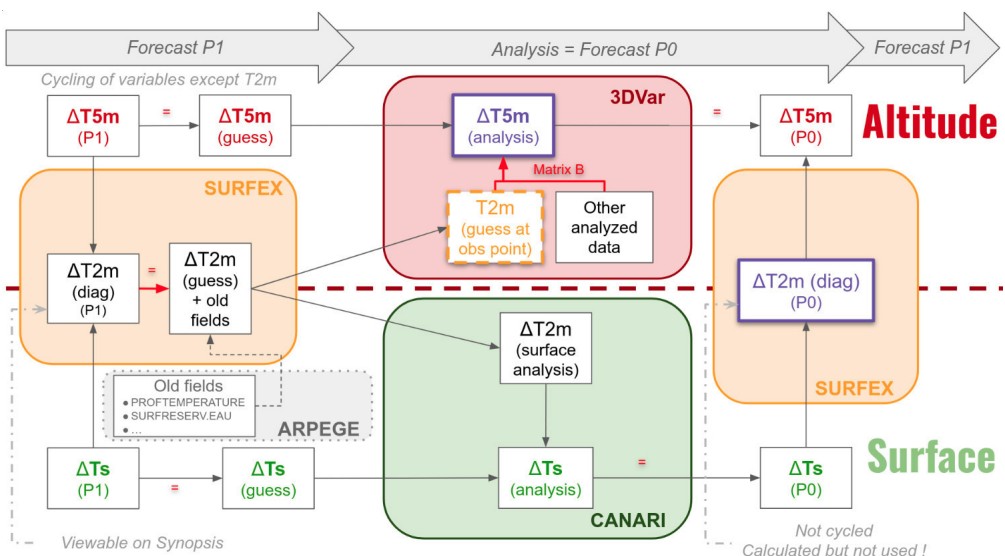

**Figure 1.** Workflow of the near-surface air temperature assimilation in Arome, featuring the altitude assimilation system (above the red dotted line) and surface assimilation system (below the red dotted line). "diag" refers to a diagnostic variable, P1 to the first term of a forecast and P0 to the initial state prior to a forecast and after the analysis step. The color boxes highlight specifically the altitude 3DVar analysis scheme, the surface Canari analysis scheme, and the diagnostics performed for T2m in the surface scheme, SURFEX.





### 2.1.2 The Canari OI surface assimilation

For the surface, the analyse is computed by the Canari system (Code d'Analyse Nécessaire à Arpege pour ses Rejets et son Initialisation) (green box of Figure 1) using the Optimal Interpolation (OI) method described by Taillefer (2009):

$$x_a = x_b + \mathbf{B}\mathcal{H}^T[\mathcal{H}\mathbf{B}\mathcal{H}^T + \mathbf{R}]^{-1}[y_o - \mathcal{H}x_b],$$

where $x_a$ corresponds to the analyzed state of the model.

Firstly, as with altitude assimilation, a quality control process eliminates observations considered to be unrealistic. For this stage, the same equation is used, but the control parameters do not have the same value. It therefore sometimes happens that certain observations are rejected in the altitude assimilation and kept in the surface assimilation. The fundamental assumption of the OI method is that very few observations are important in determining the analysis increment. So, unlike 3DVar, the
observations deemed strategic are interpolated at the grid point by a so-called structure function which models the background error covariances, i.e. $\mathbf{B}$. In our study as in operational Arome, the Mescan (contraction of MESoscale analysis and Canari) option (Mahfouf et al., 2007; Van Hyfte, 2021) activates this function which uses a correlation length of 100 km varying according to the difference in altitude between the grid point and the observations. Thus, using 2D optimal interpolation and the Mescan structure function, the analyzed temperature and relative humidity fields at 2m are obtained (box entitled "ΔTs
(analysis)" in Figure 1). The T2m and Hu2m increments calculated in the 2D canary step are then used to to compute the surface analysis, i.e. the surface temperature, average soil temperature, surface soil humidity and average soil humidity (Giard and Bazile, 2000) at each point using 1D IO.

## 2.2 Study area and in-situ data

### 2.2.1 Domain and time period

The study focuses on the alpine massifs (Figure 2: map on the right) as the mountain range having the highest number of meteorological observations and the most complex relief in France. In winter, the biases of T2m are particularly noticeable (Paci et al., 2016; Vionnet et al., 2016; Dombrowski-Etchevers et al., 2017). The experimental period ranges from 2020 to 2023 and therefore covers four winters (December, January and February): the winters 2019-2020 (with December missing), 2020-2021, 2021-2022 and 2022-2023.

### 2.2.2 In-situ data

This study makes use of the Météo-France operational observation network and of well-instrumented sites operated for research purposes (Figure 2), described hereafter. In particular, due to the mountain and assimilation focuses of the study, the operational stations used are those located in the Alps and its foothills (Pre-Alps) and taken into account by altitude 3DVar assimilation system of Arome.


**Well instrumented mountain sites**



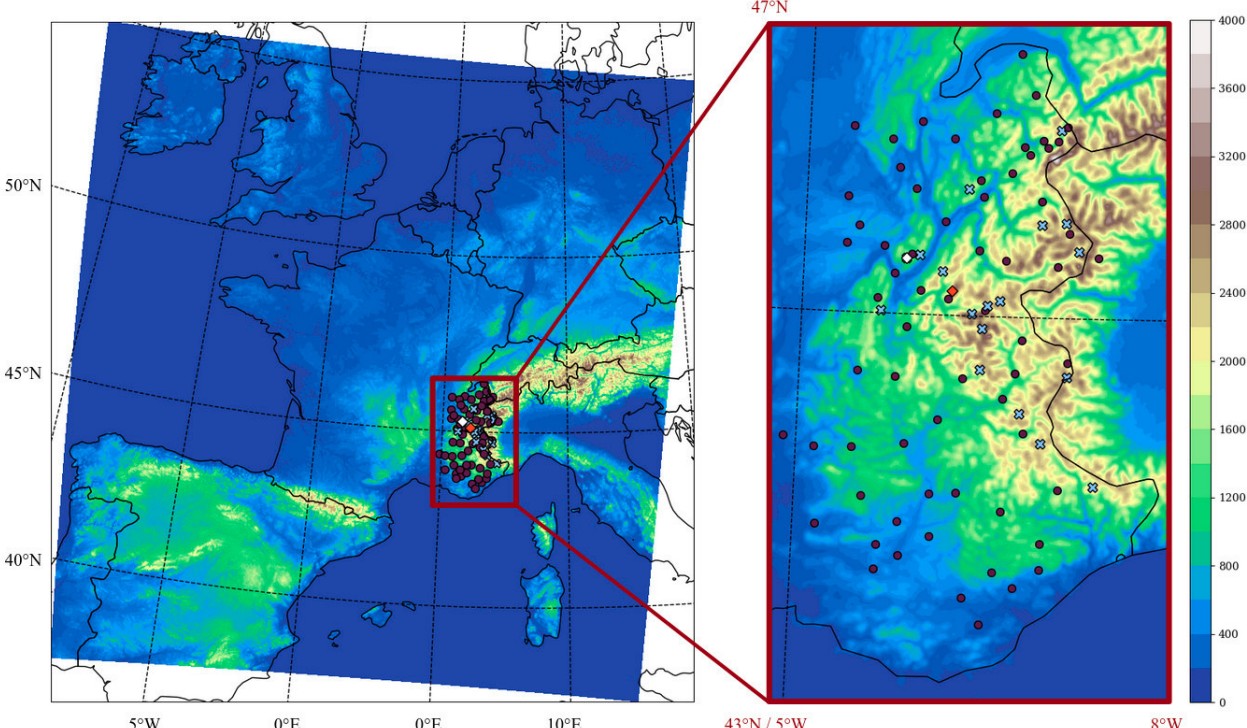

**Figure 2.** Relief of the model over the Arome-France domain with a zoom on the study domain and the measurement stations. The stations of the Météo-France standard network are shown in purple, those of the Météo-France Nivose network in blue, except for the Col de Porte-Nivose in white due to co-location with a well instrumented site, and the instrumented station at the Col du Lac Blanc in red.

- **The mid-altitude Col de Porte site (CDP, 1325m)**

  The Col de Porte (here after CDP) is an observation site located at 1325m (white dot in the Figure 2). Several variables are measured there (Morin et al., 2012; Lejeune et al., 2019), including surface and near-surface air temperatures, the latter being measured approximately between 1.5m and 2m above the surface. During the snow season the height of this temperature sensor is adjusted manually above snow surface at weekly intervals. We will consider this observation as a temperature at 2m in this paper. Besides, a Nivose station is located on the same instrumental site, measuring the temperature at approximately 5m in winter.

- **The high-altitude Col du Lac Blanc site (CBL, 2720m)**

  The Col du Lac Blanc (here after CLB) is an experimental site located at 2720m (red dot in the Figure 2) and originally dedicated to the study of wind-induced snow transport (Guyomarc'h et al., 2019; Vionnet et al., 2013; Naaim-Bouvet and Truche, 2013). The site houses various measuring instruments, one of which is a mast equipped with temperature and humidity sensors located at 2, 3.2, 5 and 7m above the bare ground. Incoming and outgoing radiation and snow





depth are also measured at the site, with for the latter a measurement co-located with every other observation due to the
high spatial variability of the snow height at this site. The snow depth provides information on the exact height of each
sensor above the surface, enabling the temperature at 2 and 5m above ground level to be retrieved by linear interpolation.
Snow was present during the study period in winter. Assuming that its emissivity is 0.98 (Dozier and Warren, 1982), the
surface temperature can be calculated from the outgoing longwave radiation by inverting Stefan-Boltzmann's law.

**Meteo-France surface observation network**

– **Standard stations**

By 'standard stations' we designate the stations from the RADOME network, that encompasses automatic stations pro-
viding hourly surface data to Météo-France (shown in purple in the Figure 2).

– **Nivose stations**

Within the RADOME network, some stations are specifically designed for high-altitude areas. They are called Nivose
stations and are mainly located above 2000m in the main massifs of metropolitan France (blue dots on Figure 2; Figure
3). They measure wind, temperature and humidity at a height of more than 2m above bare ground in order to provide
data despite a deep snowpack in winter. Generally the temperature sensors are placed at about 7m above the bare ground,
with +/- 0.5m variability depending on site configuration.

## 2.3 Numerical assimilation experiments

A good initial state is mandatory for accurate weather forecasting. In the present paper we examine the assimilation of mountain
near-surface temperatures as a possible cause for the cold bias observed in Arome forecasts. Targeted numerical experiments
are carried out by modifying the observations assimilated or the conditions in which they are assimilated in complex terrain, in
order to analyse the effect of geographical or measurements inhomogeneities on the assimilation. These numerical simulations
are be compared to a reference. This reference is the operational Arome forecast and also the one evaluated in the present study
when scores and biases are mentioned without further specification.

– **Arome-OPER.** The objective of this reference (OPER for operational version of Arome described in 2.1) is to identify
and quantify the Ts, T2m and T5m biases, be they due to the assimilation or to the modeling of processes in mountainous
terrain. The forecasts are extracted from the daily 00h run of the study period, the guess (orange-bordered box entitled
"T2m (guess at obs point)" in Figure 1) and analysis (blue-bordered box entitled "T2m (guess at obs point)" in Figure 1)
of T2m are retrieved from the 3DVar at each hour and the analysed temperatures (purple-bordered boxes entitled "T2m
(diag)(P0)" and "Δ T5m (analysis)" of Figure 1) come from the hourly analysis file.

– **NO_VALLEY.** In this numerical assimilation experiment, observations of T2m and relative humidity at 2m (RHU2m)
below 1100m a.s.l. are blacklisted before entering the 3DVar. The goal is to quantify the impact of valley stations on
assimilation in higher-altitude areas. The value of the 1100m threshold is set so that this experiment does not take into





account the data supplied by stations located in the highest VALLEYs the French Alps, such as the Chamonix-Mont
       Blanc valley with an automatic station at 1042m a.s.l. The results of this experiment will be studied over the winter of
       2022-2023.

   – **NO_NIGHT.** The diurnal cycle influences the T2m bias, which peaks at night in mountainous areas (Vionnet et al.,
       2016; Dombrowski-Etchevers et al., 2017). The Austrian version of Arome, operated by Geosphere Austria, does not
use assimilation over night. This raises the question of the impact of night-time data assimilation on the French Arome.
       To quantify this, in this "NO_NIGHT" experiment, T2m and RHU2m are not assimilated at night, i.e. when the solar
       angle is less than 10°. The impact of NO_NIGHT is being evaluated for the winter of 2022-2023.

   – **150M.** In mountains, the difference between the actual altitude and the model altitude can vary significantly. For example,
       the Mont Blanc is at 4318m for Arome 1.3 km, compared with 4809m in reality. Currently, no criteria on altitude
mismatch between model grid-point and observation station is applied to T2m assimilation in Arome. However, Quéno
       et al. (2016), Vionnet et al. (2016) and Dombrowski-Etchevers et al. (2017) considered the observations to be relevant to
       evaluate model performances and calculate scores as long as this vertical distance was less than or equal to 150m. This
       criterion was chosen as it corresponds to a 1 °C difference when considering a standard atmospheric gradient of 6.5 °C
       per vertical kilometer. In this "150M" experiment, we apply this 150m threshold and do not assimilate station data when
their altitude differs from more than 150m from their grid-point altitude in the Arome model. As a result, 13 stations are
       not assimilated. This numerical simulation is analysed for the winter of 2022-2023.

## 2.4 Scores

In the present study we use scores to quantify the agreement of model results to in-situ observations. In these scores, and
to quantify the impact of ill-suited relief, the stations which present more than 150m altitude difference with their model
grid-point are by default not discarded. The following scores will be used:

   – an hourly mean **Bias** defined as follow:

$$Bias = \frac{1}{N} \sum_{n=1}^{N} (X_n - X_{obs})$$

       where $N$ is the total number of stations and days during the studied period.

   – a root mean square error or **RMSE** which calculates an average magnitude of differences between predicted and observed
       values:

$$RMSE = \sqrt{\frac{1}{N} \sum_{n=1}^{N} (X_n - X_{obs})^2}$$

       where $N$ is the total number of stations and time-steps during the studied period.

As the RMSE alone doesn't show if a simulation is too warm or too cold compared to reality, the RMSE will be studied in
conjunction with the bias. These calculations will be done over the all study period.



In addition, the scores were also computed (1) by altitude bands: those below 1100m, those between 1100m and 2000m and those above 2000m to distinguish between valley, mid-altitude and high-altitude mountain measurements, as atmospheric conditions vary according to altitude (e.g. Chow et al., 2013; Whiteman, 2000) and Arome exhibits different biases between the valleys and the high-altitude (Vionnet et al., 2016; Dombrowski-Etchevers et al., 2017; Monteiro et al., 2022).

## 3  Results

### 3.1    Role of the inhomogeneities in sensor heights above the surface: 2m versus 5m

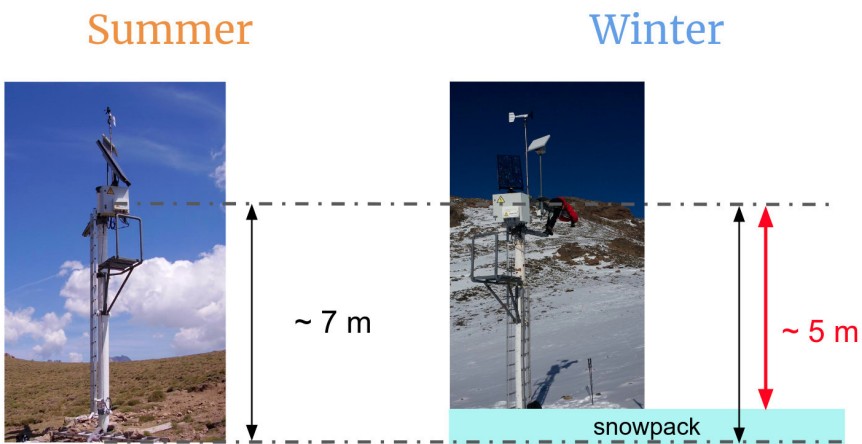

**Figure 3.** A Nivose station in summer and winter (here the Sponde Nivose station, Albertacce, Corsica). The temperature sensor (top dotted line) is located about 7m above the bare ground (bottom dotted line). Assuming for this exemple that the average height of the snowpack is 2m, the sensor is located about 5m above the surface (red arrow) in winter.

Unlike standard stations of the Meteo-France surface observation network, the Nivose stations have their sensors positioned at about 7m above the bare ground (2.2.2), so that they can provide data even in the presence of a deep snow cover. As a result of this height above snow-free ground, the temperature sensor at the Nivose in Sponde, Corsica (upper dashed line in the Figure 3) for instance records on average a temperature at 5m above the surface in winter, rather than at 2m. As stated
in the Introduction, this height specificity inherent to the Nivose stations is not accounted for when operational or research scores are calculated, for instance across different altitude bands. This introduces biases and inhomogeneities in the evaluation of modeled temperatures. Furthermore, this specificity is also not taken into account in the assimilation workflow in Arome, where the Nivose stations are not differentiated from the standard stations. Consequently, Nivoses observations are assimilated as 2m above surface observations.

This section first examines the comparability between temperatures observed at 2 and 5m above the surface at the well instrumented sites in winter. We then scrutinize how both temperatures compare in the Arome model world and with respect





to observations. Finally, we derive the impact of the erroneous comparisons between the model and the observations, both in terms of scores and with respect to the data assimilation process.

### 3.1.1 Comparison between observed T2m and T5m for CDP et CLB stations

The Arome model considers Nivose temperatures like observations at 2m above the surface. The observations at CDP and CLB help verify whether this approximation holds. We make use of DJF (December, January, February) observations for the winters 2019-2020, 2020-2021 and 2021-2022. For the first winter the month of December is missing.

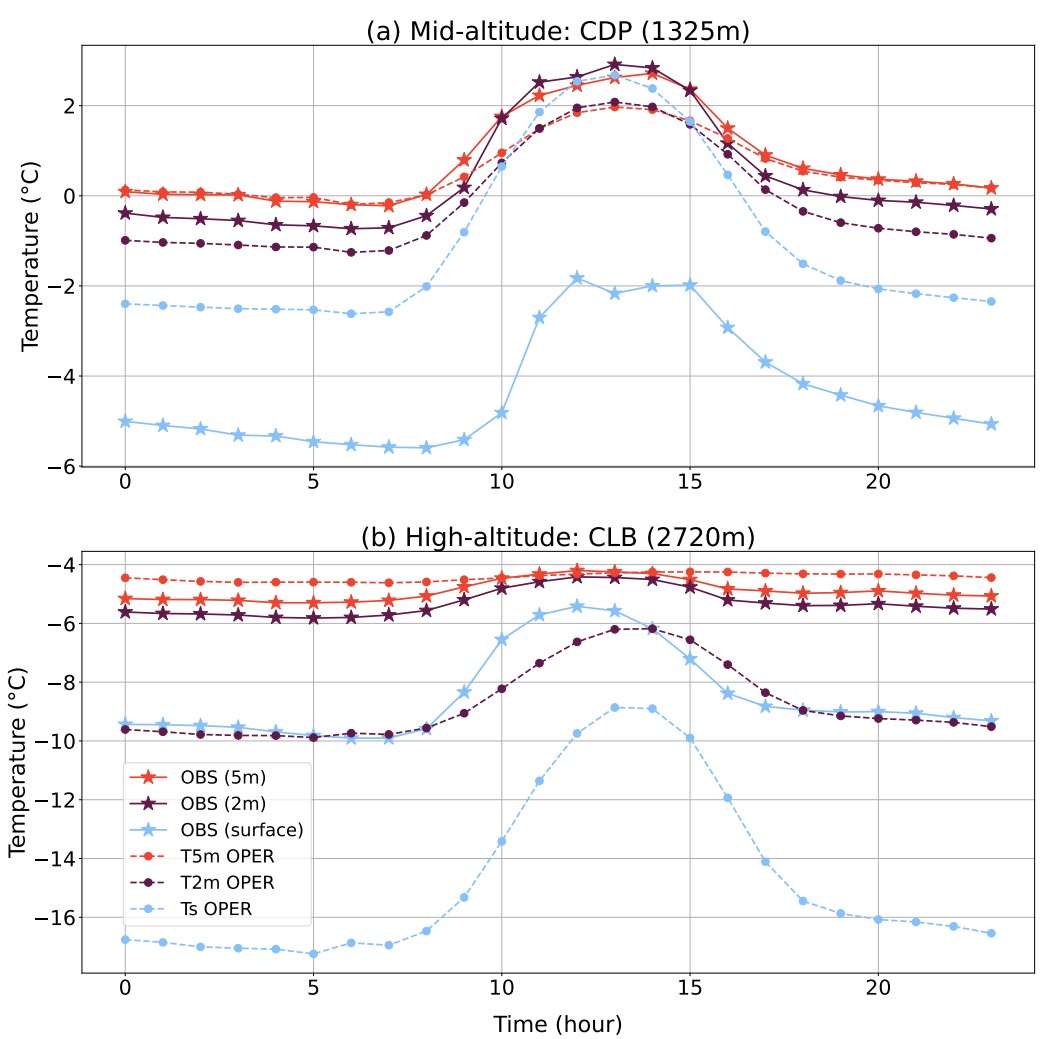

**Figure 4.** Diurnal cycle of the surface (blue), 2m (violet) and 5m (red) observed (stars with full line) and modeled (dots with dashed line) temperatures averaged over the winters of the 2020-2022 period at the CDP (1325m) (a) and at the CLB (2720m) (b).





The Figures 4 (a) et 4 (b) feature the diurnal cycles of temperatures retrieved for the surface, and at 2m and 5m, at the CDP and CLB sites. The same diurnal cycles obtained in the Arome-OPER forecasts are also shown and will be analyzed later. We

observed a mean difference between observed T2m and T5m of 0.3 °C (resp. 0.4 °C) at CDP (resp. CLB). Such a difference is not significant at CLB with respect to the measurement error, which is expertly estimated to about 0.5 °C based on the numerous co-located temperature measurements and different designs of shelters (Guyomarc'h et al., 2019). Despite a higher accuracy for the T2m observation at CDP, estimated by Morin et al. (2012) to within 0.1 °C, the T5m Nivose measurement have a lower accuracy similar to the one estimated at CLB.

Although the mean values are similar, the daily cycles of T5m and T2m observations significantly differ with maximum difference of 0.6 °C (resp 0.5 °C) at 9h at CDP (resp. at 05h at CLB) (Figure 4 (a) and (b)). In addition, the root-mean-square difference between observed T5m and T2m over winter is also significant with a value of 0.6 °C at both sites.

We note that the difference between T2m and Ts is significantly more marked than the one between T2m and T5m in the

observations, with an average difference of 4.8 °C at the CDP and 3.2 °C at the CLB; the maximum difference amounts to 6.5 °C at 10 a.m. and 4.2 °C at 7 a.m. respectively at CDP and CLB.

The diurnal cycles of Figure 4 enable to estimate mean differences between observed T5m and T2m. However the gradient between the temperatures at these two levels can be much higher than the mean values, especially during stable conditions when stratified cold air covers the Alps. December 19 and 20, 2021 typically feature this kind of situations (Figure 5).

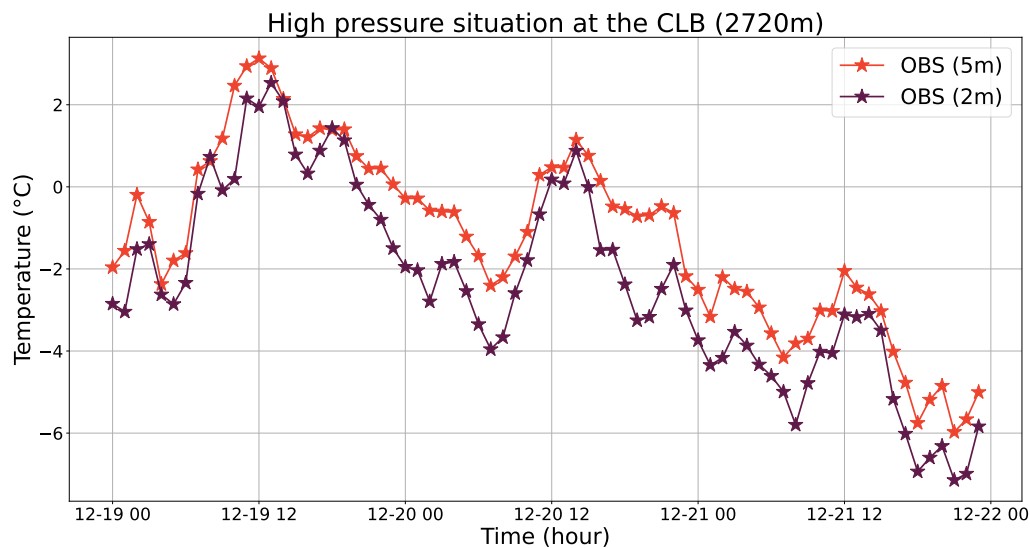

**Figure 5.** Temporal evolution of temperatures observed at 2m (purple) and 5m (red) from 19 December 2021 at midnight to 21 December 2021 at 11pm at the Col du Lac Blanc.

During this period, the Alpine massif is under the influence of an anticyclone centered on North-West Europe and reaching up to 1040 hPa. Close to the surface, the winds are weak and from the east. Despite cloudy weather on the plains, the Alps are,





on the other hand, under clear skies. During this period, the sun sets around 5 p.m. for the summits of Grandes Rousses, the massif where the CLB is located. In these stable winter conditions, the nocturnal radiation and the snow then present on the ground induce a very strong inversion in the low atmosphere (Pepin and Kidd, 2006). This induces a marked gradient between T5m and T2m reached at CLB, of up to 2.5 °C at 7 p.m. on December 20.


Thus, the approximation is invalidated: the difference between T2m and T5m is negligible on average over a winter, but is not so during certain meteorological situations. Indeed, in anticyclonic weather, particularly at night with clear skies, this difference can be greater than 2 °C and therefore very significant. Consequently, when using the observations at 5m of the Nivose stations as if they were at 2m, an error is introduced into the calculation of the scores used to qualify operational forecasts and their improvements, and also presented by Vionnet et al. (2016) and Dombrowski-Etchevers et al. (2017).


### 3.1.2 Comparison between forecasted T2m and T5m for CDP and CLB stations

Figures 4 (a) and 4 (b) also show the diurnal cycles of T5m_mod and T2m_mod simulated by Arome at CDP and CLB. The difference between these cycles is significant, with a mean gap of 0.7 °C (resp. 4.3 °C) at CDP (resp. CLB), and a maximum difference of 1.1 °C at 02h (resp. 5.3 °C at 05h). The average difference between the modeled temperatures is therefore much larger than between the observed temperatures (133% at CDP and 975% at CLB). Moreover, the gradient between the T5m_mod and at T2m_mod, is significantly stronger at the CLB than at the CDP.


Furthermore, if we compare the quality of the model at 5m and 2m, over the three winters of our study, Arome slightly overestimated the T5m at CLB with an average bias of 0.5 °C (Table 1), and underestimated it at CDP during the day with a minimum difference between the diurnal cycles of the observation and the model of -0.8 °C at 10h (Figure 4 (a)). Besides, the RMSEs of both sites have a similar value, higher than the bias. Although the model is good on average at 5m, Arome suffers from a bias in certain weather situations. Thus, the maximum error of T5m_mod over the 3 winters falls to -5.1 °C (resp. -5.2 °C) at CDP (resp. CLB), and reaches a maximum of 8.7 °C (resp.6.3 °C) (Table 1).


On the other hand, the model is too cold at 2m at both sites, with an average bias of -0.6 °C (resp. -3.4 °C) at CDP (resp. CLB) (Table 1). As with T5m_mod, the mean value of the CDP does not reflect the dispersion of T2m_mod, which ranges from -7.9 °C to 6.5 °C (Table 1). The bias and RMSE are worst at 2m than at 5m, particularly in high-altitude. Arome therefore models temperature better at its first level than at 2m.


To conclude, the T5m_mod and the T2m_mod cannot be considered as equivalent and approximated by each other for Arome. As a result, the height of the sensor should be taken into account when the observation is compared with the model.


### 3.1.3 Theoretical effects of differences between T2m and T5m in the 3Dvar assimilation

Due to the climatological differences between the observed T5m and T2m (see section 1.3.1), an error is induced during the 3Dvar assimilation if an observed T5m is considered to be at 2m, as currently done for Nivoses. Indeed, during assimilation, the guess in T2m - that is to say the diagnosis at 2m of the 1h-lead time forecast temperature - is compared to the observation





| Sites | T5m | | | | T2m | | | | Ts | | | |
| | BIAS | | | RMSE | BIAS | | | RMSE | BIAS | | | RMSE |
| | *Mean* | *Min* | *Max* | | *Mean* | *Min* | *Max* | | *Mean* | *Min* | *Max* | |
| CDP | -0.1 | -5.1 | 8.7 | 1.7 | -0.6 | -7.9 | 6.5 | 1.8 | 3.3 | -4.7 | 15.5 | 5.0 |
| CLB | 0.5 | -5.2 | 6.3 | 1.3 | -3.4 | -12.4 | 4.2 | 4.7 | -6.2 | -20.2 | 8.2 | 7.7 |

**Table 1.** Scores of Arome-OPER at CDP and CLB over the winters (DJF) between 1 January 2020 and 28 February 2022.

at 5m. The increment is then reported to the first level of the model by an observation operator (working as an adjoint to the diagnostic), and is finally used to calculate the analyzed T5m (Figure 1) (note however that within Aroma this adjoint operator is not activated). However, our analysis showed significant differences, on average and in particular in certain situations, with T5m climatologically warmer than T2m. This difference leads to a positive average bias in the analysis increment, which is on average overestimated by the assimilation of an observed T5m considered as if it were a T2m.

In addition, Arome itself has different biases at 2m and 5m, with confusion between model temperatures at these heights also having an impact on assimilation. At 2m, the model is, on average, clearly too cold for stations located above 1600m altitude (Figure 8), a bias which induces a clearly positive analysis increment in the assimilation. At 5m, on the other hand, the model has a very slight negative bias, leading to little or no analysis increment.

Thus, our analysis reveals that Arome's temperature bias further reinforces the error made by using T2m_mod instead of

T5m_mod in the analysis of snow observations. Not only is the height of the model temperature taken into account incorrect, leading to a bias linked to the climatology of temperatures at 2m and 5m in the mountains (see above), but Arome's cold bias at 2m also reinforces this error and adds an additional overestimation on average to the analysis increment. A comparison between the CLB and CDP shows that this error is greater at CLB than at CDP, due to a steeper modeled temperature gradient at the CLB (see above). Further analysis shows that, generally speaking, the T5m-T2m gradient increases with the relief Arome,

leading to greater errors at altitude, where most of the Nivoses are located.

As a result, theoretically, treating Nivose observations as measurements at 2m introduces on average a warm bias into the analysis calculated at 2m at the station, resulting from the error in the height chosen for the model temperature, and reinforced by the negative bias of the model at this height, which is much greater in magnitude than the bias of the first level of the model.

### 3.1.4 Direct verification in the assimilation system

On the basis of the CDP and CLB, we hypothesised in subsection 3.1.3 from the Results section, that the error in taking into account the height of Nivose measurements, combined with the model's cold bias of T2m, led to an overestimation of the increment. Is this validated by experiments ?

To verify this, we examine the effect of the analysis of surface observations on the analysed T5m. The graphs in Figure 6 therefore distinguish between the guess used to calculate the surface observation increment (orange line), the guess at 5m



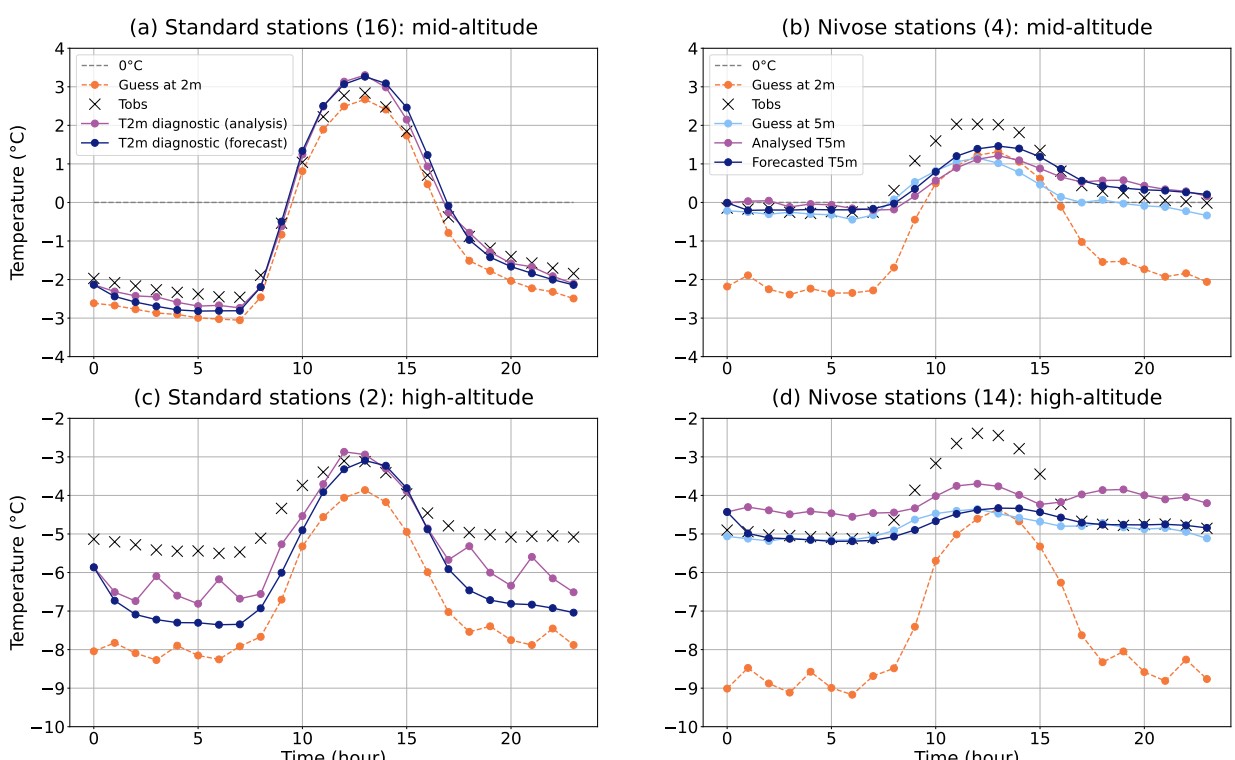

**Figure 6.** Diurnal cycles of temperature observed (crosses) or calculated at different steps within the assimilation workflow of Arome-OPER for mid-altitude mountain stations (a,b) and high-altitude mountain stations (c,d). Within each altitude range a distinction is made between standard stations (a,c) and Nivose stations (b,d); the number of stations is given in brackets. Within the modeled temperature, the guess at 2m (orange) refers to the Arome guess interpolated at the observation point (dashed orange box in Figure 1); the guess at 5m (light blue) designates the value of the guess at the first level of the model at the closest grid point of the station; the forecast temperature by the 00h run (navy blue line) also at the closest grid point of the station and the analyzed T5m (or resp. T2m diagnostic, analysis) in purple refers to the T5m analysis (or its associated T2m diagnostic, resp.) accounting for all observations incuding surface and satellite ones. The REF is the Arome-OPER forecast described in section 2.3.

(light blue line) and the temperature in the analysis (purple line), compared with the forecast temperature by the 00h run (navy blue line), by separating the Nivoses from the standard stations and also the mid-altitude (between 1000 and 2000m a.s.l.) and high-altitude mountains (at 2000m a.s.l. or more) for which the model's biases are different.

First, the guess used in 3DVar (orange line) appears too cold compared to the observations, particularly at the Nivoses. This is not surprising, since on the one hand we are comparing observations at 5m with a guess at 2m, and on the other hand the observed temperature is, on average, warmer at 5m than at 2m, i.e. at the Nivose stations than at the standard stations for

the same altitude. In addition, the guess is colder in the high-altitude mountains than in the mid-altitude mountains. This is





consistent with the model's bias at 2m, which is more pronounced in the high-altitude than in the mid-altitude mountains and which we observed in part 3.1.

Secondly, we note that the analysed T5m is worse at Nivose stations (Figure 6 (b,d)). Indeed, at high altitude, the analysed
temperature at the first level of the model shows a warm bias at night of up to 0.9 °C at 7pm (purple line in Figure 6 (d)), whereas the model is not biased at 5m. This error is potentially due to the assimilation of other observations, but above all to the height of the Nivoses not being taken into account. In fact, the value of the Nivoses increment would be much lower at night (from 6 p.m. to 7 a.m.) if it was calculated correctly using the guess at the first level of the model (light blue line) with a mean of +0.1 °C, compare to +3.8 °C with the guess at 2m (orange line) currently used. During the day at high altitude, the
forecast temperature (blue line in Figure 6) at 5m, as at 2m, is too cold with a maximum diurnal bias of -2.0 °C at 12h for the Nivoses and -1.8 °C at 08h for the standard stations.

In mid-altitude mountain areas, the analysed T5m is also worse at Nivose stations at night with a negative effect due to an overestimated increment of +2.0 °C (resp. +0.1 °C) for the guess at 2m (resp. at 5m) at night, as shown in Figure 6 (b). However, the effects are less marked because the T5m-T2m difference is smaller (cf 3.1.3). In addition, the mid-altitude mountains are
more influenced by the standard stations than high-altitude where stations are few and mostly Nivôses. The effect of height error is therefore more limited in mid-altitude mountains.

In conclusion, the assimilation of T2m seems to degrade the diurnal cycle of the analysed T5m at the point closest to the Nivoses, esp. at night. At standard stations, the assimilation generally leads to an improvement in the temperature forecast at night without any deterioration during the day (Figure 6 (b) and (d)).

## 3.2 Role of the geographic inhomogeneities of assimilated observations

In the mountains, in addition to errors linked to the poor processing of measurements from Nivose stations, other problems are present within Arome's altitude assimilation. Mountainous areas are complex to instrument and model. For some mountain stations, this results in a significant discrepancy between the model relief at the nearest point and the actual relief. What's more, the distribution of observations in mountain ranges is not homogeneous, since weather stations are mainly concentrated in the
valleys. However, in 3DVar, correlation lengths do not depend on relief, so this geographical inhomogeneity is not taken into account. These two problems are present in Arome-OPER, but not in 150M and NO_VALLEY respectively. By comparing the improvement brought about by assimilation with respect to its guess (i.e. the analysis increment) between the Arome-OPER and the experiments, we can quantify the impact of these problems, particularly in relation to the contribution of upper-air and surface observations. Although the experiments do not make any difference in their assimilation between station types, in order
to be able to compare their analysis increments with the observations, we will separate the Nivoses from the standard stations in our results.





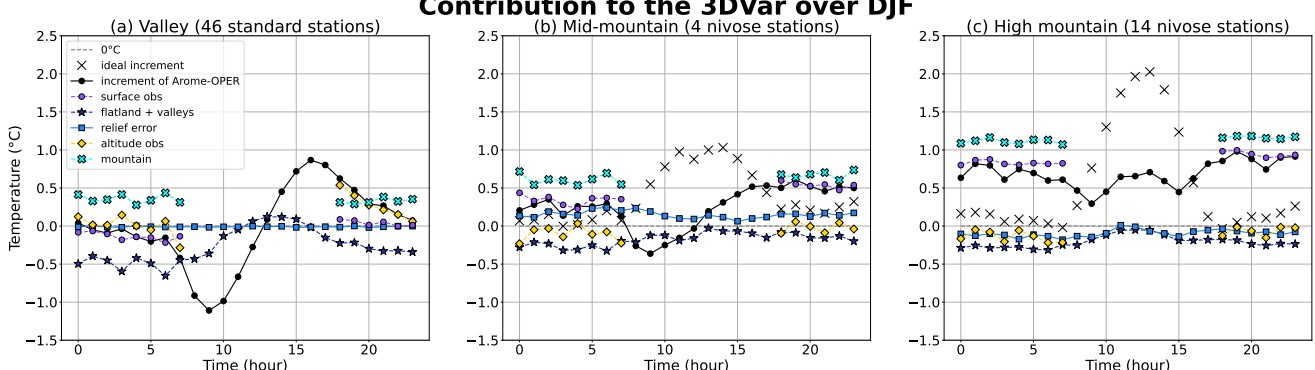

**Figure 7.** The lines in the graph represent the analysis increments obtained in valleys (a), mid-altitude mountains (b) and high-altitude, taking into account only the values from Nivose stations in mountain (b,c). The Arome-OPER analysis increment ( plain black line) is obtained by comparing the analysed T5m with its guess. The NO_NIGHT experiment enables to isolate, over night, the effect of the assimilation of altitude observations only ("altitude obs", yellow line), and, by difference to Arome-OPER, the effect of all surface observations ("surface obs", violet line). The NO_VALLEY experiment allows us to isolate the effect of flatland and valley surface stations by difference to Arome-OPER ("flatland + valleys", dark blue line). The 150M experiment enables to analyze the effect of stations with relief mismatch between model and real station location, by difference to the REF ('relief error', blue line). For the night period, we further estimate the effect of all mountain surface observation ('mountain', cyan line) by subtracting to Arome-OPER increment the increments for altitude observations and valley and flatland stations, assuming a perfect additivity of these effects and negligible mixed effects. The difference between the observation and the guess of Arome-OPER represents an idealised increment (black crosses). There is no measure at 5m in valleys, so no idealised increment is calculated.

### 3.2.1 Quantifying the impact of altitude difference between stations and model grid

In our dataset, there are 13 weather stations (out of 82) for which the model relief (of the grid point containing the station) differs by more than 150m from the station's actual altitude. These 13 stations are therefore not assimilated in the NO_VALLEY experiment presented in section 2.3.

Figure 7 shows a weak effect of relief errors on assimilation for high-altitude areas: the analysis increment calculated without surface stations impacted by an error of more than 150m between model relief and station altitude (experiment 150M), differs by only a few tenths of a degree from the analysis increment including all surface stations, which is not significant in relation to the observation error. This result is repeated at low and medium altitudes. Our results also confirm that in valleys, this error has no impact, but this is not surprising since only one station out of 46 has it (not-shown).

Stations with unrealistic relief represent a small proportion of mountain observations (15%), which explains this small effect. In fact, there are 2 Nivose stations out of 4 in the medium-altitude mountains and 2 out of 14 at high altitude. There are more standard stations with this difference in relief, with 7 out of 16 in mid-altitude mountains and 1 out of 2 at high altitudes.

Even if we focus on the diurnal cycle of assimilation at the station, the difference between Arome-OPER and 150M remains negligible and has a sign that varies and is decorrelated from the altitude. Furthermore, this difference does not depend on the





sign and value of the difference between the model relief and reality. Thus, although significant differences can occasionally be observed between 150M and Arome-OPER, stations with unrealistic terrain have a negligible impact on assimilation.

### 3.2.2    Quantifying the impact of valley stations

The disadvantage of 3Dvar is that it does not take into account relief. Valley stations therefore influence the analysis calculated
for mid-altitude and high-altitude mountains, and vice versa. By comparing the analysis increments between the NO_VALLEY and Arome-OPER experiments, we can quantify the impact of lowland and valley stations on the analysed T2m in the mountains.

We can see that lowland and valley stations cool the assimilation at all altitudes. Their impact is significant in the Alpine valleys, where night-time cooling averages -0.4 °C with a minimum of -0.7 °C at 06h (Figure 7 (a)). In the mid-altitude
mountains, their impact is weaker with a contribution of -0.3 °C at 06h. The assimilation of valley stations cools also the high altitude by -0.3 °C on average (navy blue line in Figure 7 (c)) at night at Nivoses.

At night, only the upper-air observations are assimilated by the 3DVar in NO_NIGHT. By comparing the analysis increment of Arome-OPER with the increment of this experiment, we obtain the overall night-time contribution of surface observations (purple line, Figure 7). Over night, we can hence quantify the impact of surface observations and the influence of plains and
valleys (through the NO_VALLEY experiment). By assuming a perfect additivity of assimilation effects and negligible mixed effects, we can deduce the contribution of mountain observations (from mid- and high altitudes) to assimilation (Figure 7). On average, their contribution to assimilation amounts to +0.3 °C in the valleys (cyan line in Figure 7 (a)). It hence appears that the nighttime cooling effect of valley stations on the analysed t5m in mountains, has the same magnitude than the nighttime warming effect of mountain stations assimilation onto the t5m in valleys.

## 4    Discussion

### 4.1    Impact of mountain, surface and altitude observations on assimilation

The analysis of the contribution of mountain observations shows that at mid-altitudes, mountain observations warm the assimilation at Nivose stations by 0.6 °C (Figure 7 (b)). At high altitudes, this warming is greater, with a mean contribution of 1.1 °C (Figure 7 (c)). Mountain stations therefore warm the analysis, whatever the altitude of the stations, even in the valleys.
In high-altitudes and to a lesser extent, in mid-altitudes, this warming over night degrades the performance of Arome-OPER, as illustrated by a distinctively positive assimilation increment while the ideal increment is close to zero. As the Arome-OPER T5m forecast bias is very weak at night at high-altitudes (Figure 6, we deduce that the positive increment of the analysis comes in part from the comparison of the (colder) model T2m diagnostic with the Nivose observations taken at about 5m over the surface. Another source of error is the direct use of the temperature increment at 2m to modify the model temperature at 5m,
without a transfer to the correct height-above-the-surface to calculate the analysed temperature at the first level of the model. These two influences have not yet been individually quantified. The first problem is included in the mountain contribution





(cyan line, Figure 7) while the second affects all surface observations. Finally, as mentioned in section 3.2.2, we note that the contribution of valley stations to the analysis increment in mountains, dampens the warming effect of the assimilation of mountain stations, by about 0.3 °C. This negative contribution is an artefact of the 3DVar system that does not account for relief 445 in the spatial area of influence of the increments, but has the effect of limiting the warm bias of the analysed t5m at Nivose stations. Therefore it can be seen as a compensation error within the model.

The contribution of the surface observations is by construction the composition of the mountain and valleys contributions. At night, surface observations cool the valleys by -0.1 °C on average, as the negative contribution of valley observations is higher in magnitude than the positive contribution of mountain observations (Figure 7 (a)). This helps reduce the warm bias of Arome 450 in valleys (Figure 8). The effect of surface observations is opposite in mid- and high-altitude mountains, where they warm the t5m analysis more significantly (Figure 7 (b) and (c)) due to a high positive and dominant contribution of mountain stations over valley stations. In mid-altitude areas, the contribution of surface observations reaches +0.6 °C at 18h at Nivose stations. At high altitudes, the contribution of surface observations is higher, with an average nighttime contribution of 1.0 °C at Nivose stations. These positive analysis increments are in line with the height-above-surface and missing adjoint issues mentioned 455 above.

Conversely, altitude observations warm the valleys by an average of 0.1 °C, with a maximum of 0.5 °C at 18h and cool the mountains by -0.1 °C. The assimilation of T5m is therefore mainly influenced by surface observations throughout all altitudes: from valleys to high-mountains.

## 4.2 Reviewing scores and revisiting model biases

In section 3.1.1, the importance of taking the height of the temperature sensor into account when comparing observations and the model in mountainous regions was demonstrated. The T2m scores of Vionnet et al. (2016), Dombrowski-Etchevers et al. (2017) and Quéno (2013) at high Alpine or Pyrenean stations should therefore be put into perspective. On the other hand, the 10m temperature scores of Gouttevin et al. (2023) are relevant. It is interesting to note that at 10m, the Arome model has little temperature bias, according to Gouttevin et al. (2023). What about at the first level of the model, i.e. around 5m? What is 465 Arome's "true" bias of temperature at 5m? Bias and STDE scores were calculated for stations above 2500m altitude (Nivose stations only) by comparing the T2m diagnosed by Arome and the temperature observed at the Nivoses between November 2022 and May 2023, according to the method used by Vionnet et al. (2016). On the other hand, the same scores were obtained using the T5m_mod forecasted by Arome. The stations with an altitude difference of more than 150m from their model grid point have been removed from the calculation of these scores. Indeed, although the standard vertical temperature gradient of 470 -0.65 °C / 100m is often applied to account for this difference between model relief and real relief, this is not the correct solution (Sheridan et al., 2018). In the mountains, the altimeter temperature gradient is rarely equal to -0.65 °C / 100m: it can be null, in the case of isothermics conditions with snow precipitation, positive in the case of inversions, or strongly negative. The Table 2 shows the results obtained for the old and the new evaluations.

The cold bias decreases by 2 degrees, while the STDE decreases by one degree only by comparing temperatures at an 475 equivalent height above ground level. It therefore has a significant impact on scores to evaluate models in relation to comparable





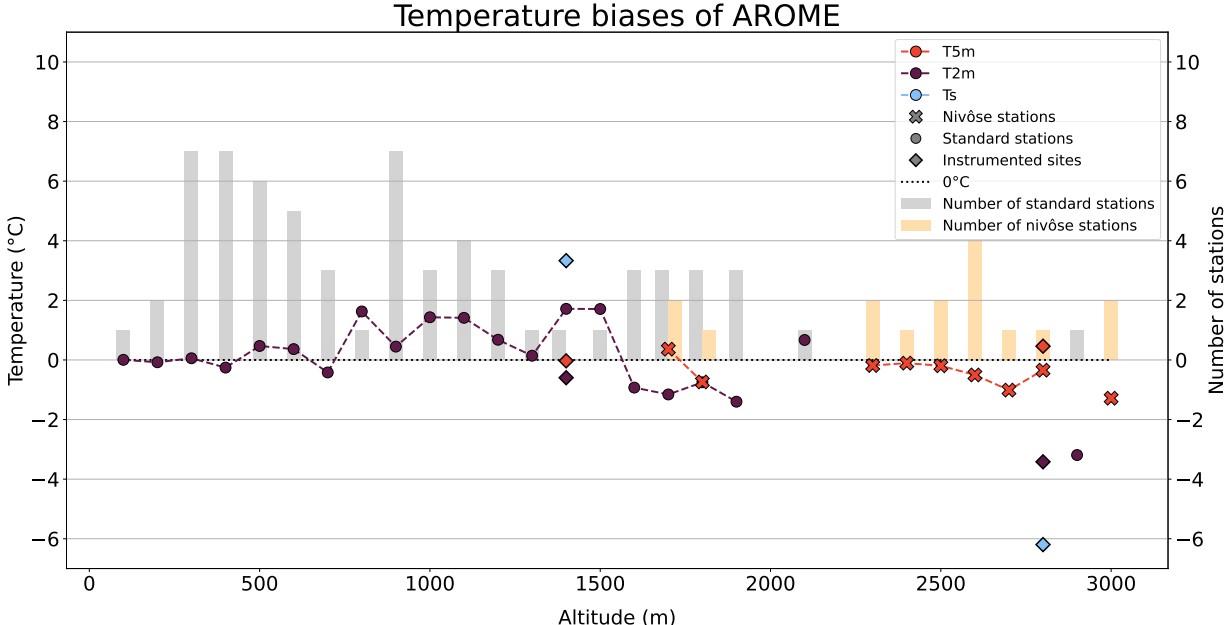

**Figure 8.** Arome-OPER temperature biases at 5m (red dashed line) and 2m (violet dashed line) at Nivôse stations (crosses), standard stations (dots) and instrumented sites (diamonds) over the winters of the 2020-2022 period. Bias is calculated by grouping stations by 100m altitude band and type. The altitude range shown in the Figure, e.g. 600m, corresponds to stations with an altitude between 500m and 600m. The number of stations used to calculate the biases is indicated by bars with the Nivoses in orange and the standard stations in grey. The Col de Porte station is counted here as an instrumented site, not a Nivose.

| | 1500m -2500m | | > 2500m | |
|---|---|---|---|---|
| | *Biais* | *STDE* | *Bais* | *STDE* |
| Vionnet et al. (2016) Method | -3.1 | 4.5 | -3.6 | 5.0 |
| Revised scores Method | -0.5 | 1.8 | -0.7 | 2.0 |

**Table 2.** Bias and STDE for Nivose stations according Vionnet et al. (2016) method (comparison with T2m_mod) and new method (comparison with T5m_mod

observations, and to bear in mind their representativeness. Taking sensor height into account has a greater impact on scores than applying a (potentially false) altitude correction. The monthly temperature bias at high-altitude Nivoses calculated for 4 months (January, April, August and November) by Dombrowski-Etchevers et al. (2017) et al has been recalculated for the period 2022-2023, on the one hand using T2m_diag (as initially) (see Figure 9a) and on the other using T5m_mod (see Figure 9b). Monthly
biases are much lower when comparing with T5m_mod than with T2m_diag, as expected. It's no longer the months with snow on the ground that are the most biased, but the months with the most solar radiation. The nocturnal bias is virtually null (slightly



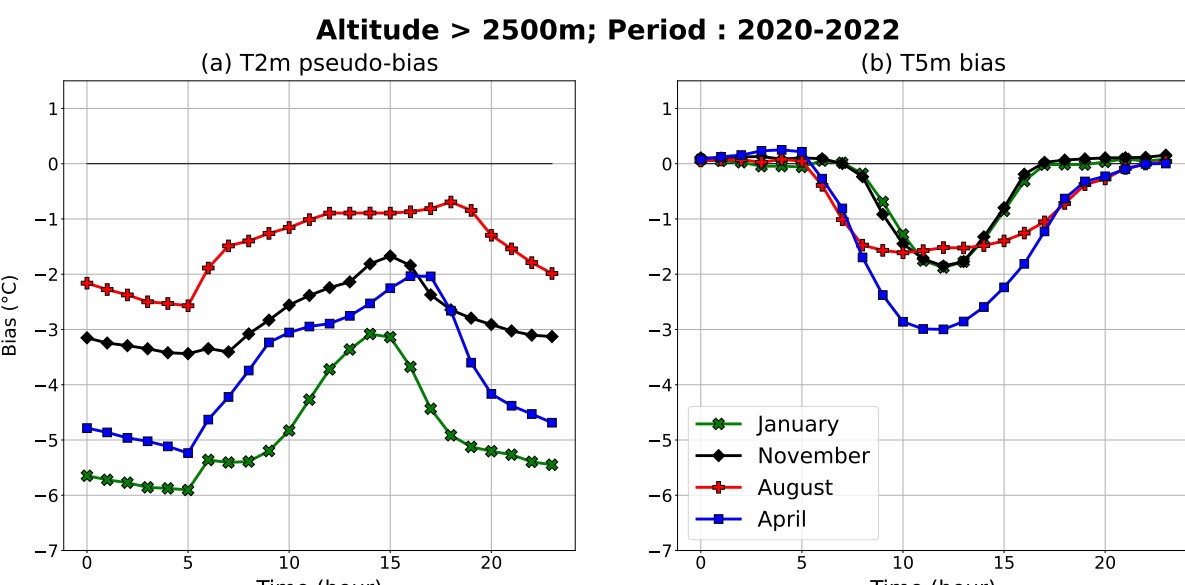

**Figure 9.** Diurnal cycle of T2m pseudo-bias as in Dombrowski-Etchevers et al. (2017) (a) and T5m bias (b) for the altitude band above 2,500m for four months of the 2020-2022 period: January, April, August and November. 6 stations meet the 150m criterion at this altitude; all are Nivose stations.

| Sites | T5m | | T2m | | Ts | |
|---|---|---|---|---|---|---|
| | *Obs* | *OPER* | *Obs* | *OPER* | *Obs* | *OPER* |
| Col de Porte | 2.9 | 2.2 | 3.6 | 3.3 | 3.8 | 5.3 |
| Col du Lac Blanc | 1.1 | 0.4 | 1.4 | 3.7 | 4.5 | 8.4 |

**Table 3.** Thermal amplitude of temperature observed and modeled by Arome-OPER at CDP and CLB over the winters (DJF) between 1 January 2020 and 28 February 2022

positive), while the diurnal bias is negative. The graph showing T5m, T2m and Ts predicted by the model versus observations confirms what was highlighted in CLB and CLP Figure 4. T5m has virtually no diurnal cycle, unlike Ts. During the day, it would appear that there is a decoupling between the ground surface and the air at the first level of the model. In addition,

the month of April is undoubtedly the most biased due to the excessive presence of snow on the ground in the model (a bias mentioned by Monteiro (2020)), which further limits the heating of the atmosphere by the surface.

    The thermal amplitude of the diurnal cycles for the 3 observed and forecast temperatures is reported in the Table 3, for both CDP and CLB. At CDP, simulated amplitudes are relatively close to observations, with a slight underestimation for T5m and T2m. However, for Ts, the thermal amplitude is overestimated. At CLB, the amplitude for T5m is underestimated. On the other





hand, it is overestimated for Ts and consequently for T2m. Arome attenuates the diurnal cycle too much at the first level of the atmosphere and accentuates it too much at the surface. This is the "decoupling" mentioned earlier.

The recalculated biases for standard stations and Nivoses are summarized by altitude in the Figure 8. Finally, the Arome model is less biased at high altitude than previously estimated. It is therefore one of the least biased models according to the synthesis by Rudisill et al. (2024) and is close to GEM-LAM evaluated by Vionnet et al. (2015). The conclusion is also quite

similar for Arome: "0.5 C Cold bias at high elevations" (Rudisill et al., 2024; Vionnet et al., 2015). It would be interesting to study the behavior of Arome at 500m vertical resolution and at 120 vertical levels (first level at around 2.5m) put into operation a few weeks ago over a domain covering the French Alps.

### 4.3   T2m diagnostic ill-adapted to complex terrain?

In the valleys and mid-altitudes, the warm bias is still present on the Figure 8. The temperature at 2m is computed (diagnosed)

from T5m and Ts. This diagnostic (Geleyn, 1988) is suitable for the boundary layer on the plains, but does not seem to be adapted to mountainous areas. It was developed at a time when the resolution of Arome's new coupler model, Arpege (Bubnová et al., 1995), was of the order of 20km over the Alps. Relief was therefore less important in the model. Meier et al. (2021) question the T2m diagnosis in mountainous areas and in the presence of snow.

According to Serafin et al. (2018) and Arduini (2017), the boundary layer is complex in the mountains. In the valley, under

a high-pressure system in winter, the wind is weak. Cold air is trapped (cold pool), so the vertical temperature profile shows a strong inversion extending vertically over several tens of meters and lasting throughout the day. At high altitude, however, the inversion will be strong in the very first few meters. It may be less than 2m thick. The diagram 10 explains this situation. In this case, the T2m diagnosic has a cold bias.

When a disturbance arrives, the cold air will remain trapped in the valleys for several hours, while the high altitude will

be under the effect of synoptic-scale circulation. The processes are therefore different for near-surface temperatures between valleys, mid-altitude mountains and high-altitude areas. T2m diagnostic must therefore be adapted accordingly.

To overcome such issues and especially solve the problem of nighttime disconnection frequently encountered between surface and atmospheric models, a prognostic surface boundary layer scheme has been proposed by Masson and Seity (2009): Canopy. This scheme was shown to foster large improvements during stable, nighttime conditions and in mountain areas, where analyt-

ical lows and interpolation methods for the temperature profile frequently fail. Thus, Meier, 2022 proposed to activate Canopy (Masson and Seity, 2009) in a version of Arome with 1.3km resolution and 90 vertical levels that runs on the Austrian domain, but adding a parameter (Inversion Factor noted IFAC) that depends on the position on the relief (plain, valley, mountain). As a reminder, Canopy was deactivated in the current version of Arome-France because a warm T2m bias was present, no doubt due to error compensation. The results are promising. Dian and Masek (2016) has proposed a modification to this T2m diagnostic,

but it is only suitable for stable or anticyclonic cases. For their part, Ingleby et al. (2024) has also proposed a revision of T2m diagnostics in the IFS model that leads to a more realistic evolution of T2m in stably stratified conditions.





## High mountain

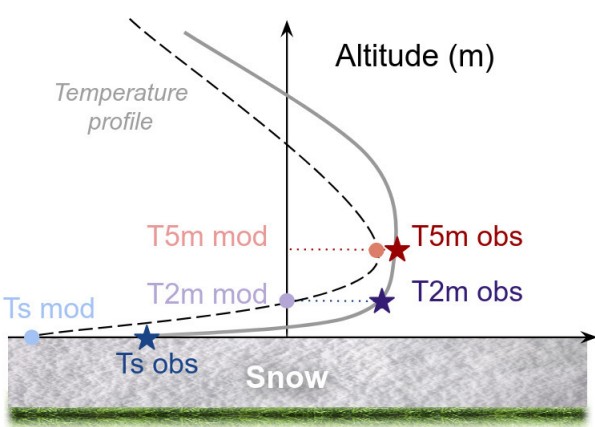

**Figure 10.** This diagram illustrates the difference between the model temperature profile (black dotted line) and the actual profile (grey solid line) at high altitude in winter. Ts is in blue, T2m in purple and T5m in red.

### 4.4 A questionable simulated Ts surface temperature?

The Figures 4 (a) and 4 (b) show a different behavior of the modeled Ts according to altitude. For example, in the high altitude of CLB, where the surface is rock which may be covered with snow, the Ts is too cold by several degrees (up to -7 °C negative
bias at night) in winter, and the diurnal cycle is too marked. This too-cold bias has an direct impact on the diagnostic of T2m. On the other hand, at CDP, in the mid-altitude region, where the surface is grassy with a forest environment, the Ts modeled is too warm (up to +4 °C positive bias during the day), while the diurnal cycle remains too marked. This is confirmed by the thermal amplitude of the modelled Ts, which is too high compared with observations in the Table 3. As a reminder, the ground scheme currently coupled to Arome is Isba-3L with D95 for the snow scheme. Etchevers (2000) had pointed out that winter
night-time cooling at the surface of the snowpack was too great in the D95 scheme (Douville et al., 1995). Similarly, Gouttevin et al. (2023) and Monteiro et al. (2022) have shown that the Ts is too cold in mountain regions. One avenue of research is to replace the Isba-3L (a force-restore scheme, with a single soil–vegetation–snow surface temperature relationship) that works together with the D95 single-layer snow scheme, by Isba-DIFF (a multi-layers soil and surface scheme that allows a resolution of specific energy balances for the soil-vegetation system as described in Monteiro et al. (2024)) associated with
Isba-ES (Boone and Etchevers, 2001), a multi-layers snow scheme. These changes that promote a more physical representation of the soil-snow-atmosphere continuum, have been tested by Monteiro et al. (2024) with modifications to improve snow cover. Finally, it would be interesting to have Ts measurements in Alpine valleys, as this warm bias could partly explain the T2m warm bias observed in valleys in winter and the model's difficulty in predicting cold pools.



# 5    Conclusions

Our study presents the impacts of inhomogeneities in the surface observation network in complex, mountainous alpine terrain, on the evaluation of the performances of a numerical weather prediction system and on the assimilation of these data themselves. The inhomogeneities studied are of three flavors: (i) the difference of altitude between the individual observation stations and the model grid-point they are located in (ii) the difference in height-above-surface of the temperature sensors across altitudes, in link with the development of thick snowpacks and (iii) the inhomogeneity in station densities between valleys and

mountain tops.

We find that the relief mismatch between stations and the model has no significant impact either in assimilation nor in model evaluation. This conclusion may be relative to the configuration of stations where this mismatch is observed in the present case study, where only 15% of stations present an important mismatch. It should be verified in other mountain regions and in particular, it may not stand for regions with more abrupt relief and more intense altitude variations like the Himalayas.

We further find that the various height-above-surface of the measurements involved across altitudes, matters. First, per se as significant differences exist in a number of meteorological situations where temperatures differ between 2m height and further up above the surface. Second, because the NWP or atmospheric models may present quite different biases at different heights above the surface, even within a few meters. In the example of the AROME system, we found that these differences, driven in part by a surface temperature bias, theoretically lead to a systematic overestimation of the assimilation increment when

temperatures actually measured at 5m above the snow surface, are considered as at 2m by the assimilation system. This is consistent with the targeted assimilation experiments that we performed, though the precise attribution of an excessive assimilation increment to the assimilation of Nivose observations alone, is not directly possible due to likely error compensations between the assimilation of other observations (esp. altitude, satellite observations) in the altitude assimilation. Most significantly, not considering the height-above-surface difference between standard and Nivose stations, has long impeded a proper quantifica-

tion and understanding of the near-surface temperature bias of Arome, probably partly hindering its resolution. Our distinction between 5m and 2m across the observational network enables to state that the temperature at the lowest model prognostic level, close to 5m above the surface, is only very moderately biased in Arome. However, temperature proves highly biased at levels below, be it at 2m above the surface or more intensely, directly at the surface itself. This makes the T2m temperature (diagnostic field) bias as much of a concern for surface modelers than for atmospheric ones. This generalizes the findings of

Gouttevin et al. (2023) based on a 2-site study in the French Alps. It also points towards the need for improved diagnostics for T2m, a variable key to a lot of applications and more widely used than the less biased T5m. Work by Dian and Masek (2016) and Meier et al. (2021) propose alternate solutions, via revisited diagnostics in stable conditions or an efficient sub-grid turbulence scheme for the lowermost atmosphere. Further work is also needed on the understanding of the turbulent exchange over snow under stable conditions, and recent or ongoing work (Stiperski et al., 2019; Stiperski and Calaf, 2023; McCandless

et al., 2022) may lead to interesting breakthroughs in the coming years in this domain. We also note that activating the adjoint of the diagnostic for the assimilation, and accounting for the proper height of Nivose measurements, should reduce the current errors in the assimilation process.



Finally, the inhomogeneities of station densities across altitudes, should definitely be considered in operational model eval-
uation procedures as biases significantly differ across altitudes, as shown by several authors (Rudisill et al., 2024; Vionnet
et al., 2016; Quéno et al., 2016; Monteiro et al., 2022) before us and reinforced by our study. Only such kind of altitudinally
differenciated evaluation can foster a better understanding of the model limitations and promote efficient model improvements
over mountain regions. With regards to assimilation, our study shows that the valleys and lower altitude stations that are the
most numerous and hence the most assimilated in the French Alps, on average have a much lower influence on the analysis of
temperature at high-altitude areas, than the high-altitude stations themselves, at least at night, for the period where we could
highlight this influence through our targeted experiments (Figure 7). This is reassuring for the Arome model with respect to the
different model biases that exist in the different altitudes, with a warm bias at low altitudes and a cold bias at higher altitudes
for T2m. This means that these differentiated biases do not majorly affect the analysis increment used for high-altitude regions,
probably due to a sufficient number of high-altitude stations. Again, this conclusion is not general and should be revisited in
other regions or even for specific regions of the French Alps with lower station densities. The use of 3DEnVar assimilation in
the new version of Arome-OPER should also limit the consequences of the inhomogeneous density of stations across altitudes
(Brousseau et al., 2024).

Our study hence defines pathways to improve the near-surface air temperature forecasting in Arome in mountain regions.
Correcting T2m biases would also enable the model's physical parameterisations to be improved. For example, Arome-OPER
does not take into account the impact in the infrared and diffuse solar of the reduction in the fraction of visible sky in the
mountains (Gouttevin et al., 2023), as this heats up the valleys and therefore exacerbates the current bias. It first appears
paramount to reduce the surface temperature bias that drives a temperature bias in the lower levels of the atmosphere, and
recent work by Gouttevin et al. (2023) and Monteiro et al. (2024) propose directions to that goal, via improved atmospheric
forcings especially for winds and radiation, and via the use of the Isba-ES-DIFF surface scheme (Boone and Etchevers, 2001).
Turbulent coupling could also be underestimated over snow-covered surfaces in the current setup of the coupling to the surface
in Arome (Gouttevin et al., 2023), a topic that could be revisited along with the ongoing scientific progress in that field. We
add as a note that katabatic winds are likely misrepresented in Arome due to a too coarse resolution in the lower atmosphere,
i.e. within the few meters above the surface. The likely underestimation of their strength provides an interesting explanatory
mechanism for the near-surface cold bias at high altitudes and warm bias in valleys, a hypothesis that we suggest here but that
has yet to be verified against in-situ data. Second, the diagnostic currently used to estimate T2m_mod from Ts and T5m_mod
is not suited for snow-covered areas in complex terrain and should be revisited. This action could also benefit from the ongoing
progress in turbulence understanding in complex terrain, and be an opportunity to elaborate a consistent adjoint for the T2m
diagnostic, to be used in assimilation. Third, we recommend the consideration and proper handling of the height-above-surface
of temperature observations, be it in model evaluation or in assimilation, as not doing so introduces evaluation or assimilation
biases. We could quantify that the bias introduced in the model evaluation amounts to up to a few degrees. And fourthly, it
would be interesting to do the same study using the 500m version of Arome over the Alps, which has 120 vertical levels, so
more levels near the surface. Indeed, Antoine et al. (2023) has showed that adding levels in the lowermost layers significantly



improves fog forecasting, thanks to better a forecast of atmospheric parameters. Having a first atmospheric level at around 2.5m could therefore improve the representation of the vertical temperature profile in the first few meters of the atmosphere.

*Code availability.* The code used for the assimilation experiments in AROME-France is owned by the members of the ACCORD consortium.
This agreement allows each member of the consortium to license the shared ACCORD codes to academic institutions in their home countries for non-commercial research. Access to codes used for the figures, can be obtained by contacting the corresponding author.

*Data availability.* The main data from the Col du Lac Blanc and Col de Porte instrumented sites are available at https://doi.osug.fr/public/ (GLACIOCLIM-CLB, 2024, 2023); complementary data can be requested for the Col du Lac Blanc from hugo.merzisen@meteo.fr. Data from the Meteo-France surface observation network and from the operational Arome-FRANCE model (analyses and forecasts) are freely
available online (Météo-France, 2025). Data from the numerical experiments performed within this study can be requested from the authors.

*Code and data availability.* All computations were performed with Python software version 3.12.3 The codes and dataset of each numerical assimilation experiment are available from a Zenodo repository (Préaux, 2025). It presents guess, forecast and analysis for each experiment used in the study as well as scripts for the following tasks: performing all data preprocessing, reading the different data sources, statistical analyses, and making figures. A second zenodo repository (GLACIOCLIM-CLB, 2025) displays additional data from the CLB with the
interpolated 2m and 5m temperatures, and surface temperature computed using incoming longwave radiation.

*Author contributions.* D. P. carried out the numerical experiments, analysed the results and wrote the core of the manuscript. I. D.-E, I. G. and Y. S. contributed to the design of the numerical experiments, and helped D.P. analyse the results and write the manuscript.

*Competing interests.* The authors declare that they have no conflict of interest.

*Acknowledgements.* The authors are grateful to Camille Birman, Pierre Brousseau and Matthieu Plu for proofreading the article, to Hugo
Merzisen for his help with the CDP and CLB data and to Patrick Moll for his insights on assimilation.
    This research was supported by the French Meteorological Institute Meteo-France. The Col du Lac Blanc and Col de Porte instrumented sites are part of the OZCAR Research Infrastructure through the GLACIOCLIM Observatory. They receive financial support from OSUG, LabEx OSUG@2020 (ANR10 LABX56), Météo-France, and INRAE.



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
