# Peer review of "On the proper use of screen-level temperature measurements in weather forecasting models over mountains"

_EGUsphere, 2025_

## Referee Comment (RC2)

Title: On the proper use of temperature screen-level measurements in weather forecasting models over mountains.
Author(s): Danaé Préaux et al.
MS No.: egusphere-2025-708

**Summary**

This manuscript investigates from different perspectives how the current usage of near-surface temperature observations, usually considered measured at 2-metre, impacts model evaluation and data assimilation in mountainous regions in an operational NWP model. The authors consider three different sources of uncertainties: 1) altitude difference between model grid-point and station height; 2) difference in the height above the surface of the sensor and the one of the model, usually assumed constant at 2-metre above the ground; 3) inhomogeneities in the observation's distributions.

This is an important topic as usually point 2) and point 3) are not considered in-depth during both model evaluation and assimilation of surface observations. A better usage of these observations can potentially improve forecasting of near-surface variables and give more meaningful indications on model developments pathways.

I find some of the conclusions worth publishing as they can be a good contribution to the present literature. In particular, the impact of the difference in the height above ground between the sensor and the model in case of snow cover is of particular importance, as this is usually neglected in operational data assimilation systems. However, there are major concerns (see "General comments" below) that should be addressed before publication.

**General comments**

The manuscript should be deeply revised before resubmission. There are typos, missing parentheses and many sentences that are very difficult to read in English, which I strongly recommend rephrasing. A few examples are given in the specific comments below, but I encourage the authors to edit the text throughout. Also, in some places the tone of writing is too "colloquial", for instance see lines 464-466. While posing questions can be a good way to engage the reader, an excessive use does not align with the scientific style of a research manuscript.

From a scientific perspective, I also strongly suggest revisiting the way the scientific questions are introduced and some of the results presented. In the introduction, it is hard to clearly get the scientific questions the authors would like to answer and what is the methodology used to address them. One way could be to clearly states what hypotheses and questions relate to model evaluation, and which ones to data assimilation. The last paragraph (lines 107 – 111) has elements of it, but it should be improved. For instance, the hypothesis that "In the present paper we examine the assimilation of mountain near-surface temperatures as a possible cause for the cold bias observed in Arome forecasts. ", in Sect. 2.3, should be introduced earlier in the text, possibly in the Introduction. Sect. 3.2 should be better introduced to guide the reader that the manuscript is now moving to the "assimilation" part and analysis of the sensitivity experiments.

Regarding the logical order, some of the "Discussion" sections seems more "Results" and should be rearranged accordingly. For instance, Sect. 4.2 seems better placed together with Sect.3.1.1. Figure 8 is discussed in Sect. 3.1.3 so it should be introduced accordingly in the text.

Some parts of the Discussion and Conclusions are too tightly related to AROME. This could be ok, as this is the numerical tool the authors are using, however I strongly encourage the authors to draw some generality out of some of these statements for the wider community, for instance talking about the physical reasons to use or replace a particular scheme rather than referring to a specific scheme's name.

From the point of view of the methodology, I have a few issues with the sensitivity experiments performed and the analysis presented in Sect. 3.2 and Figure 7.
Firstly, it is hard to understand what is plotted in Figure 7: it should be better clarified when the difference between the sensitivity experiments and the control are plotted; I do not understand the reason to use a figure legend that is different from the sensitivity experiment names, which adds confusion to the reader. Secondly, it is not clear to me what the "mountain" line represents in Figure 7: the NO_NIGHT experiment already removes the assimilation of all surface observations during nighttime (at least this is the understanding from Sect. 2.3), so what is the reason to combine it with the NO_VALLEY experiment increments? Furthermore, if the aim of the authors was to check the impact of all mountain surface observations, an additional sensitivity experiment, in which all T2m observations were removed, would have been useful. The linearity assumption used to combine the increments can be hard to justify, given the high non-linearities present in NWP models. At least the authors should justify why this experiment was not performed.

Another issue is related to the analysis in Sect. 3.1.1, in particular the author's statement that the approximation T2m ~ T5m is incorrect (line 306). From their results this conclusion is a bit misleading as on average, considering the whole winter periods, they show that this approximation is valid. The authors base their conclusion on the analysis of a specific case study in Figure 5, covering only a few days. To make their conclusion statistically stronger, I think the authors should also show a diurnal cycle of temperatures (or some statistics) computed only for anticyclonic conditions or clear-sky periods.

**Specific comments**

Abstract: "rôle" → "role"

Line 28: "… they are originally designed for Rudisill et al. (2024); Gouttevin et al. (2023)." Some parentheses might be missing for the references.

Line 35: "While primarily strong over peaks and ridges, it often comes with a warm bias in valleys. " is not clear. Do you mean "…it is often associated with a warm bias in valleys"?

Line 38: I think a few more references on the literature review and/or the forecaster's reports would be useful to better justify the 3 types of biases described.

Line 44: "wood fire heating Aymoz et al. (2007)" some parentheses are missing for the reference.

Line 46-52: Not clear, could you please reformulate? Also It seems that the reference justifying the argument is put at the end (Beauvais, 2018) but should it be placed at the beginning of the paragraph?

Line 75: what are Nivose stations? Please introduce them properly.

Line 83: "have been a preoccupation for numerous modelers", not clear. Do you mean "have been an issue recognised by numerous modelers"?

Line 83: "kind of covering up the height-above- surface adjustments that we here mention. ", please reformulate this.

Line 99: "mountain" → "mountains"

Line 108: "pitfall" does not sound very scientific, do you mean "challenges"?

Line 142: "T5m_ mod refers abusively to the temperature..", not clear. What do you mean by "abusively"? Also I think there is a trailing space between T5m_ and "mod"

Line 154: "This stage eliminates observations that are considered doubtful because they come from a non-qualified source or are too far away from the design." Is not clear, please reformulate. Do you mean that differences with the first guess are larger than a certain threshold?

Line 154: "unfairly" is not a scientific term here.

Cost function Equation: Please add an equation number to the text.

Sect. 2.1.2: Why CANARI and the 2D-OI are introduced? It does not seem this is used at all in the manuscript, as the analysis focusses on 3D-Var, as far as I understood.

Line 161: "Increments are calculated for the surface observations and for the upper-air observations. Then, J is minimized using these increments." I am not sure if here "increments" is the right terminology. Do you mean the x-xb terms in the cost function equation?

Line 168-172: This is not clear. How B is defined in the 2D OI equation? What do you mean "that very few observations are important in determining the analysis increment"? Please clarify and/or reformulate.

Line 177: "1D IO" I think should be "1D OI".

Line 180: "(Figure 2: map on the right) ", please add a label to each panel and refer as "Figure 2b".

Line 204: "..., with for the latter a measurement co-located with every other observation due to the high spatial variability of the snow height at this site." It is not clear, please reformulate.

Line 213: I think the authors refer to the Nivose stations in the text before these are described/introduced. Please adjust the text.

Line 221: This hypothesis would be better placed at the end of the introduction to better illustrate the scope of the work. I think this is somehow described at line 96-97, but it is not clear enough, in particular the link to the model cold bias. The authors should also explain why an assimilation deficiency should cause a forecast bias (at which lead times?).

Line 224: "These numerical simulations are be compared to a reference. ". Please reformulate.

Figure 3 caption: "exemple" → "example"

Line 283: "et" → "and"

Line 306-310: "Thus, the approximation is invalidated: ..." please reformulate. It is not clear at this stage what approximation the authors are referring to, even though it is clarified in the remaining of the paragraph.

Line 306-310: Can this argument be better generalised for instance by computing a diurnal cycle only for the anticyclonic conditions in the considered 4-year period?

Line 343: The discussion in the text "jumps" from Figure 4/5 to Figure 8, which is a bit confusing. Please readjusts the figure order or the text.

Section 3.1.3 is difficult to follow. Could you please reformulate the Section to better distil the main message the authors would like to convey? For instance, reducing where possible references to "see above text" or if necessary, better explaining where a reader should focus, e.g. a figure or a particular subsection.

Line 369: "Secondly, we note that the analysed T5m is worse at Nivose stations". Worse than what?

Line 383: "esp. ". please correct this.

Line 387: "Mountainous areas are complex to instrument and model.", please clarify what "complex" means in this context.

Table 2: typos in "Biais", no closing parenthesis in figure caption.

Line 495-497: Please rephrase.

Line 513 – 520: This discussion is too much related to Meteo-France's models and using a lot of details that are not of interest to an external reader. The authors should

give the information that would be of interest for the community, not the namelist settings used in operational AROME.

Line 588: "Correcting T2m biases would also enable the model's physical parameterisations to be improved.". Is it not the other way around, improvements in physical parameterisations reducing the T2m biases in the first guess and hence reducing the "activity" of the data assimilation system?

Line 596: Is there any reference to previous work that could back up this hypothesis of misrepresented katabatic winds?

Line 599-606: This long paragraph mixes perspectives and hypotheses for future work with conclusions from this work. I would suggest reorganising it so that the conclusions from this work are clearly divided from the perspectives. E.g. the recommendation to properly consider the height above the surface of the observations in assimilation and evaluation should come as a conclusion, whereas reducing temperature biases through improved model physics (using Isba-ES-DIFF etc.) should come as a perspective.

---

## Author Comment (AC1)

**Response to the Referee#1 for manuscript : Preprint egusphere-2025-708**

Referees' comments are in **black.**

Authors'answers are in **blue**, *text from the original manuscript in black, italic* and *modified or added portions in blue italic.*
* * *
**Summary:**

1.1 - This study addresses an important topic related to discrepancies in measurement and modeled surface air temperature heights. The study provides evidence supporting that the height differences between surface air temperature measurements should be accounted for when evaluating model performances and assimilating data. These contributions will be valuable to publish and account for in future research and operational modeling; however, there are major revisions required prior to this paper being suitable for publication that are addressed below.

We thank the reviewers for careful reading of the manuscript and helpful suggestions.

**Overarching comments/concerns:**

1.2 - The manuscript requires thorough editing by a native English speaker prior to its resubmission. There are common wording errors, citations outside of parenthesis, grammar issues, and awkwardly worded sentences that need to be resolved prior to publication. Examples are provided in the first 5 specific comments below, but this comment applies throughout the manuscript.

We carefully revised the wording, references and grammatics of the manuscript.

1.3 - Analyses for Section 3.1 are only conducted at 2 sites. This seems lacking and would require a justification of this limitation. Why are the other stations (e.g., from Figure 6) not included in this initial analysis? Even if both 2m and 5m temperature observations are only available in a few locations, it seems this analysis can and should be broadened by: (i) comparing modelled 2m and 5m temperature across a broader spatially continuous domain, and (ii) comparing modelled data with more ground observations, and group results by station height to evaluate the potential discrepancies between simulated T2 and T5, and provide deeper insights on how validating modelled T2 with observed T5, or DA practices, can induce issues. (i) Could be further used to evaluate how discrepancies between modelled T2 and T5 vary with geographic, climate, and vegetation conditions.

The Col de Porte and Col du Lac Blanc are to the best of our knowledge the only mountain stations with observations at both at 5m and 2m above the surface in France. This indeed poses an intrinsic limitation to the observation-based evaluation of model temperatures at these height levels, especially for a joint evaluation at these 2 levels. However we do fully agree that a broader evaluation of AROME 2m and 5m temperatures, with the stations from Fig6 and as performed in Fig 8, would strengthen our conclusions and broaden their impact in this section.

We therefore added **a new subsection within section 3.1, entitled "Assessment of forecasted T2m and T5m across the Alps"**. This section shows that the differences seen at the research sites between forecasted T5m and T2m in Arome, are actually quite representative of the entire Alps, where a negative difference between T2m-T5m is a generalized pattern and grows with altitudes across the mountain range (**new Figure 6 a,** see below). We furthermore now show in this section the evaluation of Arome against all T2m and T5m observations available across the French Alps, that was previously shown in the Discussion section 5.1 (now Figure 6b in the revised manuscript, Figure 8 in the original manuscript). This figure illustrates that the model biases highlighted in Sect 3.1.2 at the research stations, are representative of the general model behaviour in the French Alps.

[Figure]

*New Figure 6 a. Arome-OPER mean temperature differences between 2 m and 5 m as a function of altitude, for each model grid-point over the study area for winter 2021-2022. Orange line denotes median, boxplots mark the 25%-75% percentiles, blue whiskers the 5%-95% percentiles and dots the values outside this range.*

1.4 - It was not clear why results were presented in the order they were presented, and it is not particularly easy to follow. A clear explanation for the paper's logical flow to start the results section, e.g., focusing on addressing specific science questions, would be very useful.

As an answer to the concern of clarity expressed by both reviewers, we thoroughly revised the structure of the manuscript. The main modifications are the following :

- First, **also as an answer to a concern expressed by Referee#2**, we made **the research questions** addressed and the hypothesis tested within the manuscript **much clearer in the Introduction**. We formulated them so that they provide the overarching structure of the manuscript, see in particular the modifications in the last paragraph of the Introduction, reported hereafter :

*"In a nutshell, the present study intends to draw the light on some pitfalls affecting the use of the near-surface air temperature observations in mountain terrain for numerical weather forecasting, through addressing a series of research questions :*

*Taking the example of the Arome-France NWP system that operationally runs over a large alpine region, we will first address the question of the impact of varied sensors' heights above surface, on the assessment of model performances. One of the underlying questions is whether observations acquired at 2m to about 5m above the snow surface, can be used without specific treatment to evaluate model performances, or whether they should be considered separately, as revelatory of different model behaviors. Through this analysis, we intend to provide guidelines for the use of temperature measurements for model evaluation in mountain regions.*

*In a second subsection of the Results, we will evaluate the effect of this height heterogeneity on the way the model is corrected by assimilation. This subsection will answer the question of whether the height of the observation above the surface matters for assimilation, or whether it is not necessary to discriminate between temperatures from 2 to 5m above the surface for the assimilation. In particular, we will examine the assimilation of mountain near-surface temperatures as a possible cause for the cold bias of Arome.*

*Finally, another question poorly addressed in existing literature, is how the relief mismatch between observation stations and model grid-cell, and valley-vs-mountain heterogeneities in terms of observational density, affect the efficiency of data assimilation. We will address this question in a final Results subsection of this study, through the use of dedicated assimilation experiments.*

*The plan of our manuscript addresses these items sequentially, after a section dedicated to material, method and study area. To the best of our knowledge these questions have not thoroughly been addressed in mid-latitude mountain regions of the world. We focus on winter conditions as the period when the model biases are the strongest. We also take the opportunity to propose in a Discussion section perspectives to circumvent the problems highlighted, for the benefit of weather forecasting in complex terrain."*

- Second, the **Results section 3** is now clearly structured into **3 subsections** dedicated to each of these 3 research sections:
    - **3.1 Impacts of heterogeneous sensors'height** (t5m vs t2m) **on model evaluations**
    - **3.2 Effects of heterogeneous sensors'height** (t5m vs t2m) **on the 3DVar assimilation**
    - **3.3 Effect of other heterogeneities within the mountain observation network** (namely : altitude differences between stations and model, and station density heterogeneity), **on the 3DVar assimilation**
- As already mentioned earlier, a new subsection was created within 3.1 to demonstrate the spatial generalization of the differences between the modelled t5m and t2m seen at research sites (see comment 1.3). This subsection also integrates a full evaluation of the t2m and t5m model biases at all stations available within the study area, showing consistent results with the evaluation performed at the 2

research sites. This general evaluation was previously part of the Discussion, and we agree with both reviewers that it rather belongs to the results section.

- **A better distinction was made between Results, Discussion and Conclusion:** as stated above, we relocated parts of the Discussion to Results, but we also relocated parts of the original conclusion, into the Results section (see the penultimate comment by Referee 1, also raised by Referee 2). We completely rewrote the Conclusion, following the 3 overarching research questions and synthesizing the main results relevant to each of them.
- Finally, we added within the Data and Methods "Assimilation experiments" subsection a new paragraph dedicated to expliciting the methods used to analyse these experiments, based on the analyses increments. The questions raised by both reviewers regarding the original Figure 7 of the manuscript, showed that an explanation of these methods would be a beneficial addition to the text.
- We are currently finalizing the revised version of the manuscript where the referees will be able to see these changes.

**Specific comments:**

L18: "Becken (2010)" citation should be inside parenthesis. The reference is now corrected.

L25: Also, at local & global scales. The sentence has been rephrased.

L28: "were" rather than "are", and citations in parenthesis. This has been corrected.

L34: "high" altitude regions. This has been corrected.

L33-36: Awkwardly worded sentences, suggest revising. The sentence has been rephrased.

L95-96: This is a crucial statement for the paper's scope and therefore requires citation(s).

It is difficult to find a citation that mentions that the specific characteristics of mountain stations, and in particular the height of the measurement, are not taken into account, because the Arome assimilation system currently treats all stations identically. This seems so obvious that it is not even mentioned. As an illustration, the recent Marimbordes et al., 2024 paper that describes the forthcoming evolutions for the surface assimilation (CANARI part) in AROME, presents in its Figure 3 the map of the 2-m temperature observation stations assimilated within CANARI. This map features stations above 2700 m altitude, all of which are actually Nivôse where air temperature is measured at about 7 m above the snow-free ground (a zoom helps notably distinguish the Ecrin-Nivôses (2970 m a.s.l.) and the La Meije-Nivôse (3100 m a.s.l) as white dots across the 45° parallel on the Figure). But the height of the measurement is not mentioned. We added this example in the revised manuscript to anchor our statement in recent scientific literature on assimilation incl. over mountain regions :

*"As an illustration, Figure 3 in Marimbordes et al. (2024) shows a map of so-called "2-m temperature observations stations that are assimilated" in the surface assimilation. This map includes high-altitude (> 3000m a.s.l.) stations from the Meteo-France "Nivôse" observation network, that measure air temperature actually at roughly 7.5 m above snow-free ground"*

Paragraph starting in L53: This paragraph seems to focus on cold biases, but biases reported as positive values. If the bias is a cold bias, then it should be reported as a negative number (i.e., model – obs). This has been corrected.

The introduction could also benefit from including the motivation of the snow-albedo feedback. That is, surface air temperature biases can propagate to snowpack biases (e.g., in snow cover) which can have albedo feedbacks due to the high albedo of snow that in turn feedback to and increase the original temperature biases.

We recognise the snow albedo feedback as an important motivation for our work and added a dedicated paragraph :

*"Several publications have pinpointed the links between snow cover and near-surface temperature (cold) biases, with the persistence of a too-extended snow cover and possible limitations in snow-atmosphere exchanges and representation of ablation processes in the models, invoked as possible sources for too cold temperatures over snow (Vautard et al., 2013 ; Kotlarski et al., 2014). In particular, near-surface air temperature is involved in the estimation of the snow-albedo feedback (Scherrer et al., 2012), a mechanism by which snow aging and/or disappearance, enhancing the surface albedo, leads to an increased absorption of solar radiation by the surface and further surface warming or melt (Peixoto and Oort, 1992). Winter et al. (2017) and Kotlarski et al. (2015) have among others highlighted the links between temperature biases in high-resolution climate models and the magnitude of this feedback, with models suffering from negative biases over snow and ice artificially overestimating the temperature response upon snow disappearance."*

Figure 1 should be presented more clearly, (e.g., with (a), (b), (c), etc) labeling to show the flow of the figure. This has been corrected.

There are many definitions and abbreviations used throughout the paper. There should be a table in Methods which clearly defines these.

We will introduce one of such tables in the revised version of the manuscript, in the Material and Method section or as an Appendix, based on the example below but with shorter descriptions in the last column :

| Category | Abbreviation | Signification |
|---|---|---|
| **MODEL** | T5m_mod | Temperature at the first level of the model (approximately 5 m) |
| | T2m_mod | Temperature diagnostic at 2 m according to Geleyn (1988) |
| | Ts_mod | Surface temperature of the ground for Arome |
| **OBSERVED** | T2m_obs | Observed temperature at 2 m, above the bare ground for standard stations and above the surface at instrumented sites CDP and CLB |
| | T5m_obs | Observed temperature at 5 m above the surface; measured at Nivose stations and at CLB |
| | Ts_obs | Observed temperature at the surface; measured at instrumented sites |
| **STATIONS** | CDP | Instrumented site of the Col du Lac Blanc, located at 2720 m |
| | CLB | Instrumented site of the Col de Porte, located at 1325 m |
| | Standard | Automatic stations providing hourly surface data to Météo-France; sensors are 2 m above the bare ground |
| | Nivose | Automatic stations designed for the mountains; sensors are 7 m above the bare ground |
| **EXPERIENCE** | OPER | Operational Arome forecast (Arome-OPER) |
| | NO_VALLEY | Numerical assimilation experiment in which observations of T2m and relative humidity at 2 m (RHU2m) below 1100 m a.s.l. are blacklisted before entering the 3DVar. |
| | NO_NIGHT | Numerical assimilation experiment in which T2m and RHU2m are not assimilated at night, i.e. when the solar angle is less than 10° |
| | 150M | Numerical assimilation experiment which do not assimilate station data when their altitude differs from more than 150 m from their grid-point altitude in the Arome mode |

**Table A1.** Key abbreviations used in the study for modeled and observed temperatures, type of stations and numerical assimilation experiments

Figure 4: It may be more useful to have OPER and OBS lines on separate panels, and show shading for respective lines to represent temporal variability.

One of the purposes of Figure 4 is to compare model behaviour (OPER) to observations, as analysed in the submitted version in section 3.1.2.

As separating OPER and OBS on different panels makes this comparison much uneasier, we preferred to stick to representing them on the same panel. However we welcome the suggestion of the reviewer to also represent variability. To do so while limiting the complexity of the figure, we now propose 3 separate panels for T5m, T2m and Ts, enabling the representation of variability as well as a comparison between model (Arome-OPER) and observations for each of these temperatures. See this new Figure 4 and its caption below.

This new Figure seems to us much easier to read, but has the drawback that it makes it difficult to compare T5m with T2m and Ts (be it observation or model) at each site, so that we propose to keep the original Figure in Supplementary.

[Figure]

**New Figure 4.** *Diurnal cycle of the 5 m (a, red), 2 m (b, violet) and surface (c, blue) observed (OBS) and modeled (OPER) temperatures averaged over the winters of the study period at the CDP and CLB research sites. The shaded (resp. hatched) areas represent the observed (resp. modelled) variability through the 25-75% percentile range.*

Can you provide an explanation for the differences between the measurement heights, particularly why max daily T2 is larger than max daily T5, but T2 is lower than T5 in most other time steps (at CDP); whereas at CLB, T5 is higher at all time steps relative to T2. Importantly, because only 2 sites are analyzed, and the sites show differences in patterns, how can results be generalizable?

The different behaviour between the mid-altitude CDP site, and the high-altitude CLB site, can be explained by the differences in dynamics of the lower atmospheric boundary layer between both topographic, meteorological and physiographic settings.

CDP is located at a mid-altitude pass surrounded by elevated mountains (> 2000 m on the eastern side). The site is furthermore located in a large meadow surrounded by ~35 m-high coniferous trees, experiences moderate wind speeds (1.4 m/s mean annual windspeed over 1993-2023) and sometimes exhibits patchy or no-snow conditions even in winter. This, in conjunction with the surrounding forest with much lower albedo than snow, enables the development of a convectively-driven mixed layer on the course of the morning on clear-sky days, whereby the 2m air temperature becomes ephemerally higher than the 5m one at midday. We insist however that the difference between t2m-obs and t5m-obs at midday is lower than 0.5°C and on the order of magnitude of measurement uncertainty.

On the other hand, CLB is a more open and higher altitude site. Because nearby relief is less present, the lower atmospheric boundary layer is more influenced by the nearby free atmosphere and less by the surface. Furthermore, larger wind speeds (mean winter wind

speed over 2006-2026 : 4.9 m/s) contribute to a shallower temperature inversion over the snow surface and reduced daily range in temperatures (Oke, 1987). The signature of this is visible in the t2m and t5m daily cycles that exhibit a much reduced amplitude w/r to what happens at CDP. The continuous snow cover around the site over the winter and the absence of surrounding surface elements subject to solar heating, contribute to maintaining a temperature inversion at least up to 5 m height and a shallower boundary layer all day long during winter.

In the end, the main differences between the 2 sites in the winter, lie in a more developed boundary layer for the mid-altitude, that manifests through different amplitudes in daily cycles (well explained by differences in wind speeds and differential heating of the surroundings), and the ephemeral crossing between t5m and t2m at midday at CDP that lies within measurement uncertainty. Otherwise, the behaviours of the lower boundary layer at CDP and CLB do not differ much. As the identified differences are in line with processes described in literature, we do not put in question their general validity. Much more different are the model behaviours at these sites, which require a deeper scrutiny developed in subsection 3.1.2.

Finally, would these discrepancies in diurnal cycles look different for periods of snow cover vs. no snow cover (e.g., winter vs. summer)?

To answer this question we here provide a figure (Figure R1) similar to the New Figure 4, but over summer months JJA:

[Figure]

***Figure R1****: Diurnal cycle of the 5 m (a, red), 2 m (b, violet) and surface (c, blue) observed (OBS) and modeled (OPER) temperatures averaged over the winters of the study period at the CDP and CLB research sites over JJA 2020-2022. The shaded (resp. hatched) areas represent the observed (resp. modelled) variability through the 25-75% percentile range.*

These discrepancies in diurnal cycles are mostly different at CDP for the surface temperature, as induced by the contrast between the presence vs absence of snow. Indeed snow is majorly present at this site in winter while completely absent in summer. While the winter surface temperature is therefore capped at 0°C, the summer surface temperature is much higher than T2m and T5m as consistent with the diurnal pattern of radiation and convection development. As the signature of these effects is already present at CDP in winter except for observed surface temperature, due to the presence of surrounding canopies considered in the model grid point, the general pattern in air temperatures above the surface and ranking between model and observations is the same as in winter, with enhanced diurnal amplitudes.

At CLB the differences between both seasons is less marked as snow is regularly present until early to mid-July at the site, making half of the summer months similar to winter in terms of snow surface conditions. The winter and summer dynamics at the site are therefore closer, though with a much weaker T2m negative bias in the model, completely disappearing at mid-day and possibly induced by the development of convection and more air mixing facilitated in snow-free or patchy snow conditions. The model biases in terms of surface temperatures remain of similar magnitude, which could be caused by numerous reasons (soil thermal and optical characteristics and their representation in the model, snow staying too long in the model as assessed by e.g. Monteiro et al., 2024) that are beyond the scope of this paper.

We took the opportunity of this discussion to clearly state in the Abstract and Introduction **the focus of the paper on the winter season when the model biases are the strongest.**

Throughout the paper I recommend using different wording than "guess" which is confusing (e.g., in Figure 6). Guess is also not clearly defined making the results related to this wording difficult to follow

We acknowledge that the term "background" is indeed more common in data assimilation, although the wording "first guess" or just "guess" is also regularly used, at least in the NWP community. All occurrences of "guess" have been replaced by "background", in the text as well as in the Figures.

L363:366: I am not sure if this makes sense, because the guess at 2m is also much lower than the diagnostic analysis and forecast at 2m as well.

In the Fig 6 of the submitted manuscript, we indeed see that the background at 2 m is colder than the analysis and forecast at 2m at standard stations. There are actually a few differences in the estimation of the background vs the analysis/forecast: first, the former is estimated based on the four nearest neighbours, while the latter are taken at the closest point to the stations. Second, the background comes from a 1h lead time forecast, launched each hour, while the forecast itself comes from the 00h run for its 24 first hours of prevision (24 first "terms" of the prevision). Both effects are responsible for the differences between guess at 2 m on the one side, and analysis/forecast at 2m on the other side (the analysis being warmer than the forecast thanks to the correction induced by the assimilation of observations).

At our 16 mid-altitude standard stations this difference is weak (0.4°C mean difference between guess and forecast) as should be expected as differences between 1h forecast vs 1-to-24h forecast, and between the 2 spatial interpolation procedures used, are usually not major. However it becomes stronger at the 2 high-altitude stations, on the order of 1°C. We think that this is primarily an effect of the 4-nearest-neighbour vs closest-point algorithm applied to extract the background vs the analysis/forecast, and also a side-effect of only 2 stations being used in this Figure for T2m at high-altitudes, namely La Masse and Mont Cenis stations (**Figure R2** below). Indeed, because so few stations are available, "local effects/configurations" have an important impact on the statistics. This is particularly striking in the case of the La Masse station (2800 m a.s.l) that exhibits a strong altitude difference with the model grid-point, making Arome not very representative of the ground-truth conditions. The nearest neighbour point in Arome is at 2548 m a.s.l while the mean of the 4 nearest neighbours are at 2506 m a.s.l.. At this station the difference between guess at 2 m, and forecast, is particularly high and we think that beyond differences in lead-times between both modelled fields, this may likely be due to differences in altitude between the neighbouring points and modelled altitudinal temperature gradients between these points. Altitude differences are lower for the Mont Cenis stations and come together with more limited difference between guess and forecast.

[Figure]

Figure R2: Diurnal cycles of temperature observed (Tobs, crosses) or calculated at different steps within the assimilation workflow of Arome-OPER for the 2 high-altitude standard mountain stations, similar to Figure 6 of the original manuscript.

We added a short explanation of this in the manuscript:

"*Note that for technical reasons, the interpolation procedure for the background temperature at 2 m involves the 4 nearest grid-points to the station, and differs from that used for the other model products (nearest model grid-point only). This induces a structural difference between the background at 2 m and for instance the forecast at 2 m, that is usually below*

*0.5°C but can be enhanced by local effects when only few stations are considered like in Figure~\ref{cycle_assim_REF}c, resulting in that case in a background at 2 m being distinctively colder than the forecast at that height.*"

Figure 8: it does not seem to make sense that the symbols should be connected with dashed lines. These results may be better presented in a table format than a figure.

We feel that the dashed lines connecting the symbols, as well as the presentation of the results in a Figure, help capturing the altitudinal evolution of the model biases at the different heights. It also makes clear how instrumented sites fill gaps in the standard observation networks, by enabling to assess biases at heights usually not scrutinized in some altitudinal ranges (e.g. the T5m bias at the mid-altitude site CDP). Therefore we stand for keeping this figure as it is, but we also propose to relocate it to Section 3.1 in accordance with the manuscript structural changes recommended by both reviewers, and put it together with another new figure (new Figure 6a, see above) showing the difference in T2m-T5m in Arome-OPER as a function of altitude, in support of the assessment of the generalized behavior of Arome T2m vs T5m across the study area.

Figure 9: pseudo-biases are not clearly defined and therefore it is difficult to make sense of this figure. This is now better explained.

Overall, much of the discussion section seems more like additional results sub sections, rather than a true discussion of the authors' perspectives on the results and insights for future research.

The Results and Discussion sections have been reorganised for greater clarity, see point 1.4.

The Conclusions section should be shortened to more concisely highlight the key takeaways and implications. Much of the discussion that is currently in the Conclusions section may be better placed in the Discussion section.

We agree with the reviewer and relocated most of the previous "Conclusion" into the Discussion section, by creating a first subsection entitled "Summary and general perspectives". We entirely rewrote the conclusion, which now concisely highlights the main findings of the study in link with the research questions.

Please make data used for this study publicly available to support reproducibility.

All data and code to analyze them were provided. We took the opportunity of this revision to better distinguish between code and data availability in the revised version of the manuscript.

---

## Author Comment (AC2)

**Response to Referee#2 for manuscript : Preprint egusphere-2025-708**

Referees' comments are in **black.**

Authors'answers are in **blue**, *text from the original manuscript in black, italic* and *modified or added portions in blue italic.*
* * *
2.1 - This manuscript investigates from different perspectives how the current usage of near-surface temperature observations, usually considered measured at 2-metre, impacts model evaluation and data assimilation in mountainous regions in an operational NWP model. The authors consider three different sources of uncertainties: 1) altitude difference between model grid-point and station height; 2) difference in the height above the surface of the sensor and the one of the model, usually assumed constant at 2-metre above the ground; 3) inhomogeneities in the observation's distributions.

This is an important topic as usually point 2) and point 3) are not considered in-depth during both model evaluation and assimilation of surface observations. A better usage of these observations can potentially improve forecasting of near-surface variables and give more meaningful indications on model developments pathways.

I find some of the conclusions worth publishing as they can be a good contribution to the present literature. In particular, the impact of the difference in the height above ground between the sensor and the model in case of snow cover is of particular importance, as this is usually neglected in operational data assimilation systems.

We thank the reviewer for this assessment.

However, there are major concerns (see "General comments" below) that should be addressed before publication.

General comments

2.2 The manuscript should be deeply revised before resubmission. There are typos, missing parentheses and many sentences that are very difficult to read in English, which I strongly recommend rephrasing. A few examples are given in the specific comments below, but I encourage the authors to edit the text throughout. Also, in some places the tone of writing is too "colloquial", for instance see lines 464-466. While posing questions can be a good way to engage the reader, an excessive use does not align with the scientific style of a research manuscript.

We thank the reviewer for pinpointing some of the language errors, edits and typos, and we are thoroughly revising the manuscript in view of the submission of a revised version.

2.3 From a scientific perspective, I also strongly suggest revisiting the way the scientific questions are introduced and some of the results presented. In the introduction, it is hard to clearly get the scientific questions the authors would like to answer and what is the methodology used to address them. One way could be to clearly states what hypotheses and questions relate to model evaluation, and which ones to data assimilation. The last

paragraph (lines 107 – 111) has elements of it, but it should be improved. For instance, the hypothesis that "In the present paper we examine the assimilation of mountain near-surface temperatures as a possible cause for the cold bias observed in Arome forecasts. ", in Sect. 2.3, should be introduced earlier in the text, possibly in the Introduction. Sect. 3.2 should be better introduced to guide the reader that the manuscript is now moving to the "assimilation" part and analysis of the sensitivity experiments.

We thoroughly revised the structure of the paper and especially the research questions, which are now much better framed in the Introduction, and formulated so that they provide the overarching structure of the Results sections. We invite the Referee to read the **answer to point 1.4 from Referee 1** for a complete answer to this comment.

2.4 Regarding the logical order, some of the "Discussion" sections seems more "Results" and should be rearranged accordingly. For instance, Sect. 4.2 seems better placed together with Sect.3.1.1. Figure 8 is discussed in Sect. 3.1.3 so it should be introduced accordingly in the text.

The Results and Discussion sections have been reorganised for greater clarity. We especially followed your advice to put section 4.2 into the Results section 3.1; it now constitutes a dedicated subsection 3.1.4. Similarly, the former Figure 8 has been relocated in the Result section 3.1.3 (see point 1.4 from Referee 1).

2.5 Some parts of the Discussion and Conclusions are too tightly related to AROME. This could be ok, as this is the numerical tool the authors are using, however I strongly encourage the authors to draw some generality out of some of these statements for the wider community, for instance talking about the physical reasons to use or replace a particular scheme rather than referring to a specific scheme's name.

This has been improved and we invite the referee to read the new version of the Discussion in the revised manuscript, as it would be too long to reproduce it here.

The Conclusion was entirely rewritten, accounting for this comment and also as demanded by Referee#1. It now follows the overarching research questions explicit in the Introduction, and is reproduced here:

*"This study investigated the impact of inhomogeneities of the observational network specific to mountain regions, on the evaluation of the NWP system Arome and on the effects of surface data assimilation within this system.*

*We first questioned whether the differences in height above the surface between sensors should be cared for when evaluating models in terms of near-surface air temperature. These differences are correlated with altitude and induced by the need to prevent the sensors from being buried in thick snowpacks in high-altitude terrain over the winter. We showed that T5m and T2m should not be considered equivalent when performing model evaluations: despite a limited mean difference over winter at our mid- and high-altitude research sites, both temperatures can differ significantly in specific situations, especially low-winds and clear skies. Therefore, taking one for another introduces errors. Furthermore, at the instance of the Arome model, atmospheric models may present very different biases at these different heights, so that the confusion between both temperatures leads to erroneous interpretations of model biases. We therefore recommend a distinct evaluation of modeled T5m and T2m against the relevant observations in mountain terrain. Only such kind of altitudinally differenciated evaluation can foster a better understanding of the model limitations and promote efficient model improvements over mountain regions.*

*We then questioned whether this difference in height plays a detrimental role in assimilation, as observations at 2 or 5 m are not discriminated within the assimilation system of Arome, an approximation that we estimate may be common among NWP systems. We showed that indeed, this confusion between heights in the assimilation process, leads in the case of Arome to an overestimation of the analysis increment in high-altitude regions, inducing an overestimation of T5m analysis at night and a degradation of performances with respect to the model without assimilation (background or forecast).*

*Finally, we questioned the effect of station vs model relief mismatch, and higher density in valley stations, onto the assimilation. The differences in altitude between stations and model grid-points, does not affect significantly the performance of assimilation. This may be due to the limited number of stations with an important (higher than 150 m) relief mismatch with respect to the Arome model, that runs with a high spatial resolution coming with a better representation of the topography that models with coarser spatial resolutions. With respect the imbalance between observation stations across altitude, we find that it also weekly affects the assimilation: as a matter of fact, the effect of low-altitude stations at high-altitude locations, is of the same order of magnitude that the effect of high-altitude stations assimilation onto the temperature of low-altitude areas. However, this effect is quite strong, changing the analysis temperature by about +/- 0.3°C. This means that data from a different altitude, bring a noticeable correction to the model at another altitude, where the biases can be different. This result illustrates the limitations of the current 3DVar assimilation system disregarding the effect of topography in the spatial structure of assimilation increments. Our analysis further confirms the strong analysis increment, at high-altitudes, induced by the assimilation of Nivose stations as if their measurements were at 2 m above the surface. We showed that this effect is probably the reason why the assimilation of surface observations degrades the performances of Arome in this altitude range, while relying on upper-air data (satellite, radar..) assimilation only would produce a better analysis.*

*To summarize, this study helped define guidelines for the improvement of high-resolution NWP systems in mountain terrains: In particular, sensors'height should be considered both in model evaluation and assimilation; topography should be accounted for in the spatial structure functions involved in assimilation; model biases at 2 m height and lower could possibly be reduced by the use of diagnostics more appropriate to mountain terrain, a higher number of vertical levels in the models and enhanced work on the surface scheme to improve the representation of soil-snow-atmosphere energy transfers."*

2.6 From the point of view of the methodology, I have a few issues with the sensitivity experiments performed and the analysis presented in Sect. 3.2 and Figure 7. Firstly, it is hard to understand what is plotted in Figure 7: it should be better clarified when the difference between the sensitivity experiments and the control are plotted; I do not understand the reason to use a figure legend that is different from the sensitivity experiment names, which adds confusion to the reader. Secondly, it is not clear to me what the "mountain" line represents in Figure 7: the NO_NIGHT experiment already removes the assimilation of all surface observations during nighttime (at least this is the understanding from Sect. 2.3), so what is the reason to combine it with the NO_VALLEY experiment increments? Furthermore, if the aim of the authors was to check the impact of all mountain surface observations, an additional sensitivity experiment, in which all T2m observations were removed, would have been useful.

The linearity assumption used to combine the increments can be hard to justify, given the high non-linearities present in NWP models. At least the authors should justify why this experiment was not performed.

To clarify what is plotted on Figure 7, we added a new section in the Material and Methods section, entitled "2.4.2 Analysis of the experiment". This section defines the analysis increments that we use to analyze the results of the experiments, and describes precisely how we combine the increments of the different experiments to estimate the effects of valley, mountains, altitude.. observations. We also distinguish between *analysis increments* ($\Delta$) and *virtual analysis increments* ( $\Delta^v$), the latter being diagnosed from complementary, denial

experiments and therefore incorporating compound effects which we cannot disentangle.
For instance, for nighttime, the NO_NIGHT experiment that suppresses the assimilation of surface temperature and humidity observations over night, enables to retrieve the contribution of upper-air (altitude) observations only. For the nighttime period, we define the virtual analysis increment from surface observations only, $\Delta^v$obs_surface, as:

$$\Delta \text{OPER} = \Delta^v\text{obs\_surface} + \Delta \text{NO\_NIGHT}$$

This virtual analysis increment for surface observations only, likely differs from the one that would have been calculated by disabling the altitude analysis, due to compound effects between altitude and surface observations. In the decomposition we propose, these compound effects are integrated in the surface observation *virtual* analysis increment, hence distinguished as *virtual*. We now clearly acknowledge that the experiments performed do not enable to quantify the compounds effects.

The increments defined in this new section 2.4.2, are incorporated in the legend of the Revised Paper Figure 7 for consistency and enhanced clarity.

The fact that the NO_NIGHT enables the analysis of the surface observation contributions *for the nighttime only*, is now stated more clearly.

Please find below the **Revised Paper Figure 7 (now Figure 9)** and the **new section 2.4.2 "Analysis of the experiments"**:

[Figure]

*Revised Paper Figure 7 (now Figure 9):* *Analysis increments (denoted Δ) obtained in different configurations of the pool of assimilated observations, as described in subsection 2.4.2. These increments are retrieved at stations'locations in valleys (a), mid-altitude mountains (b) and high-altitude, taking into account only Nivose stations in mountains (b,c). the difference between the observation and the background of Arome-OPER represents an idealised increment (black crosses). There is no measure at 5 m in valleys, so no idealised increment is calculated*

**New section 2.4.2 "Analysis of the experiments"** (see next page)**:**

**2.4.2 Analysis of the experiments**

In the Results section 3.3, the above-mentioned assimilation experiments will be analyzed to quantify the effect of varying observational network characteristics onto the assimilation result (i.e. the analysis). These characteristics include the exclusion of valley and flatland stations, of all surface stations at night, and of stations for which the altitude difference with respect to the model grid-cell exceeds 150 m. To highlight the effect of these variations in observational networks, we make use of the analysis increment $\Delta$, whereby :

$$\Delta = x_a - x_b \tag{3}$$

with $x_a$ the analyzed model state and $x_b$ the background model state prior to assimilation.

At observation stations, an ideal analysis increment would enable the analysis to fully coïncide with the observation. We therefore define the ideal analysis increment at stations as:

$$\Delta_{ideal} = x_{obs} - x_b \tag{4}$$

where $x_{obs}$ denotes the observation.

The NO_NIGHT experiment, disabling the assimilation of surface observations at night, enables to highlight the effect of the altitude observations only for the nighttime period. We hence call for the nighttime period :

$$\Delta_{obs\_altitude} = \Delta_{NO\_NIGHT} \tag{5}$$

For the nighttime period, we hence can define a virtual analysis increment coming from the analysis of surface observation only, $\Delta_{obs\_surface}^{v}$, by considering the following relationship between the analysis increment of the Arome-OPER experiment ($\Delta_{OPER}$), and the ones that respectively result from the assimilation of altitude ($\Delta_{obs\_altitude}$) and surface observations ($\Delta_{obs\_surface}^{v}$) only:

$$\Delta_{OPER} = \Delta_{obs\_surface}^{v} + \Delta_{obs\_altitude} \tag{6}$$

In practice, this virtual analysis increment for surface observations only, likely differs from the one that would have been calculated by disabling the altitude analysis, due to compound effects between altitude and surface observations. In the decomposition proposed in relation (6), these compound effects are integrated in the surface observation analysis increment $\Delta_{obs\_surface}^{v}$, hence distinguished as a *virtual* increment analysis, and we do not have the possibility to quantify them.

Similarly, the analysis increment of Arome-OPER can also be decomposed on into the virtual contribution from the flatland and valleys $\Delta_{valleys}^{v}$, and what comes from the upper-air and mountain stations only included in the NO_VALLEY experiment. According to this decomposition:

$$\Delta_{OPER} = \Delta_{valleys}^{v} + \Delta_{NO\_VALLEY} \tag{7}$$

and also:

$$\Delta_{OPER} = \Delta_{valleys}^{v} + \Delta_{mountain}^{v} + \Delta_{obs\_altitude} \tag{8}$$

where relation (7) enables to retrieve $\Delta_{valleys}^{v}$ while relation (8) enables to retrieve the contribution from mountain stations only among surface observations, $\Delta_{mountain}^{v}$.

Another possible decomposition of $\Delta_{OPER}$ reads:

$$\Delta_{OPER} = \Delta_{150M} + \Delta_{>150m}^{v} = \Delta_{obs\_altitude} + \Delta_{<150m}^{v} + \Delta_{>150m}^{v} \tag{9}$$

where $\Delta_{>150m}^{v}$ (resp. $\Delta_{<150m}^{v}$) is the virtual analysis increment for surface stations with more (resp. less) than 150 m altitude departure with respect to model relief, while $\Delta_{150M}$ refers to the 150M experiment.

In these latter relations, similarly to the $\Delta_{obs\_surface}^{v}$ increment, the virtual increments, denoted by a $v$ exponent, are not directly calculated from an experiment but diagnosed from a complementary experiment, and therefore include compounds effects that cannot be isolated.

These different increments will be used in the Results and Discussion sections to analyze the effects of heterogeneities in the observational network in Alpine terrain, on the assimilation in Arome.

2.7 Another issue is related to the analysis in Sect. 3.1.1, in particular the author's statement that the approximation T2m ~ T5m is incorrect (line 306). From their results this conclusion is a bit misleading as on average, considering the whole winter periods, they show that this approximation is valid. The approximation is valid in mean, meaning that the mean difference between T5m and T2m is within the (quite high) observational error assumed at the CLB site and at the CDP for the T5m only. But the approximation is certainly not valid when regarding diurnal amplitudes, diurnal cycles or **error scores** like the RMSE. We now say this more clearly in the manuscript.

2.8 The authors base their conclusion on the analysis of a specific case study in Figure 5, covering only a few days. To make their conclusion statistically stronger, I think the authors should also show a diurnal cycle of temperatures (or some statistics) computed only for anticyclonic conditions or clear- sky periods.

We will incorporate a statement and statistics on this aspect in the revised version of the manuscript, also referring to Gouttevin et al., 2023 who analyzed the diurnal cycles in temperatures at different heights above the surface under clear-sky vs cloudy conditions at CLB in winter. We here present an analysis performed over one winter period (DJF) at CLB, distinguishing between clear skies and cloudy skies based on an atmospheric effective emissivity criterion similar to Gouttevin et al 2023 (**Figure R3**): the 25% days with strongest (resp. lowest) effective emissivities are classified cloudy (resp. clear-sky). Additionally, we distinguish the low-wind periods within the clear-sky days, by selecting the moments when wind speed is lower than 4 m/s. Figure R3 shows that the daily cycles in temperature differences between 2m and 5m at the CLB, largely differ between cloudy conditions and clear-sky conditions. In the former there is almost no difference between t5m_obs and T2m_obs, while in the latter the mean difference is strong especially at night, reaching -0.6°C between 0 and 8 UTC. This difference is larger than the measurement uncertainty. It is even stronger in clear-sky, low-wind conditions when the mean DJF difference at night is below -0.7°C for several hours.

[Figure]

**Figure R3:** Daily cycles of the T2m_obs minus T5m_obs difference at CLB over one winter period (DJF), distinguishing the **clear-sky** and **cloudy** weather conditions. '**all-sky**' represents the mean DJF daily cycle (all-sky conditions). The low-wind periods within the clear-sky days are also distinguished (**clear and low-wind**).

Specific comments

Abstract: "rôle" à "role". This has been corrected.

Line 28: "… they are originally designed for Rudisill et al. (2024); Gouttevin et al. This has been corrected.

(2023)." Some parentheses might be missing for the references. All references have been corrected

Line 35: "While primarily strong over peaks and ridges, it often comes with a warm

bias in valleys. " is not clear. Do you mean "…it is often associated with a warm bias in valleys"? Yes, and this is now corrected.

Line 38: I think a few more references on the literature review and/or the forecaster's reports would be useful to better justify the 3 types of biases described. References to Arnould and Preaux, 2021; and Beauvais, 2018, have been added.

*Arnould, G. and Préaux, D.: Study of AROME Temperature in mountain regions, ACCORD Newsletter, 2021.*

*Beauvais, L.: Fronts chauds sur les Alpes : Hiver 2017-2018. Comportement des modèles, ateliers de la prévision du Centre-Est, 2018.*

Line 44: "wood fire heating Aymoz et al. (2007)" some parentheses are missing for the reference. This has been corrected.

Line 46-52: Not clear, could you please reformulate? Also It seems that the reference justifying the argument is put at the end (Beauvais, 2018) but should it be placed at the beginning of the paragraph? We reformulated the sentence and positioned the reference at the beginning:

*"The second Arome warm bias manifests in valleys when a warm front encounters the relief, especially in the direction perpendicular to the valleys and ridges (Beauvais, 2018). In these situations, the warm front penetrates too rapidly or too deeply in the valleys, leading to a modelled rise in temperature that is too strong and often generates an altitudinal upward shift in the snow-rain transition in the model. As a result, the model can forecast rainfall instead of snowfall in the valleys, where the major roads are."*

Line 75: what are Nivose stations? Please introduce them properly. This has been corrected:

*"This is typically the case in France, where the sensors of the high-altitude observation network for snow and mountain meteorology, the so-called "Nivose" stations, are about 7 m above the snow-free ground."*

Line 83: "have been a preoccupation for numerous modelers", not clear. Do you mean "have been an issue recognised by numerous modelers"? We rephrased following this suggestion.

Line 83: "kind of covering up the height-above- surface adjustments that we here mention. ", please reformulate this. This has been reformulated:

"*It may have until now obliterated the possible issue of height-above-surface adjustments*"

Line 99: "mountain" à "mountains" OK done

Line 108: "pitfall" does not sound very scientific, do you mean "challenges"? Yes we switched to "challenges"

Line 142: "T5m_ mod refers abusively to the temperature..", not clear. What do you mean by "abusively"? Also I think there is a trailing space between T5m_ and "mod" We suppressed tha abusively as it was misleading :

*"For convenience, in the diagram and in the rest of the article, T5m_mod will refer to the temperature at the first level of the model, which is approximately at 5~m above the surface."*

Line 154: "This stage eliminates observations that are considered doubtful because they come from a non-qualified source or are too far away from the design." Is not clear, please reformulate. Do you mean that differences with the first guess are larger than a certain threshold?

Yes, there was an error in our sentence, and we corrected "design" for "background".

Line 154: "unfairly" is not a scientific term here. We reformulated:

*"However, if this background is biased, the screening can also reject observations that come from accurate measurements and contain valuable information for the assimilation"*

Cost function Equation: Please add an equation number to the text. All the equations are now labelled.

Sect. 2.1.2: Why CANARI and the 2D-OI are introduced? It does not seem this is used at all in the manuscript, as the analysis focuses on 3D-Var, as far as I understood. Indeed the focus is on the 3DVar in terms of assimilation scheme. However, we also examine the temperature biases of AROME-oper at 2m above the surface. This temperature, T2m_mod, is the result of a diagnostic that considers both T5m_mod and Ts_mod, the latter being affected by the surface analysis CANARI. It therefore seemed necessary to introduce CANARI to present all the schemes affecting our temperatures of interest. We added this to the manuscript for more clarity.

*"The analyzed surface temperature is involved in the estimation of the analyzed temperature at 2 m via a diagnostic (Figure 2c)".*

Line 161: "Increments are calculated for the surface observations and for the upper-air observations. Then, J is minimized using these increments." I am not sure if here "increments" is the right terminology. Do you mean the x-xb terms in the cost function equation?

Indeed, *'increment'* is not the right term: what is meant is *'background departure'*. We have corrected this error in the manuscript.

*"Background departures are calculated for the surface observations and for the upper-air observations. Then, J is minimized using these background departures."*

Line 168-172: This is not clear. How B is defined in the 2D OI equation? What do you mean "that very few observations are important in determining the analysis increment"? Please clarify and/or reformulate.

In the case of CANARI, B is a static, univariate matrix (meaning that it does not integrate any inter-variables correlations, e.g. no correlation between T2m and Hu2m). It relies on a very large correlation-length of 100 km, used almost isotropically as only modulated by sea/land surface mask and topography (Marimbordes et al., 2024). The large correlation length works so that only a few observations are enough to have an impact on the analyzed state.

OI was traditionally used for its numerical efficiency and flexibility, via the possibility to reduce the system to a "small" matrix inversion, considering only a limited number of relevant observations for the correction of the model state (see Durand et al., 1993 for an early time application in data-scarce mountain areas).

We added these elements in the revised version of the manuscript and rephrase the sentence on the few observations for more clarity : "*OI is an assimilation method particularly suited in the context of rather scarce data, when a limited number of observations are used to determine the analyzed state (Durand et al., 1993).*"

*Durand, Y., Brun, E., Merindol, L., Guyomarc'h, G., Lesaffre, B., and Martin, E.: A meteorological estimation of relevant parameters for snow models, Annals of glaciology, 18, 65–71, 1993.*

Line 177: "1D IO" I think should be "1D OI". This has been corrected.

Line 180: "(Figure 2: map on the right) ", please add a label to each panel and refer as "Figure 2b". This now done in the revised manuscript.

Line 204: "…, with for the latter a measurement co-located with every other observation due to the high spatial variability of the snow height at this site." It is not clear, please reformulate. We reformulated the paragraph.

Line 213: I think the authors refer to the Nivose stations in the text before these are described/introduced. Please adjust the text. This has been adjusted.

Line 221: This hypothesis would be better placed at the end of the introduction to better illustrate the scope of the work. I think this is somehow described at line 96-97, but it is not clear enough, in particular the link to the model cold bias. The authors should also explain why an assimilation deficiency should cause a forecast bias (at which lead times?). This is now accounted for through the research questions and hypotheses stated in the Introduction, that specifically mention this hypothesis.

The effect of assimilation on the forecast clearly depends on the variables/situations/phenomena at stake, and could last for one to a few hours or even more, but it is hard to precisely tell this in the present context without a dedicated evaluation. We changed the formulation as the question of effects on the forecasts was actually not addressed in our study:

" *In particular, we will examine the assimilation of mountain near-surface temperatures as a possible cause for the cold bias of Arome*."

Line 224: "These numerical simulations are be compared to a reference. ". Please reformulate. We corrected the typo: are to be compared.

Figure 3 caption: "exemple" à "example"  This has been corrected.

Line 283: "et" à "and" This has been corrected.

Line 306-310: "Thus, the approximation is invalidated: …" please reformulate. It is not clear at this stage what approximation the authors are referring to, even though it is clarified in the remaining of the paragraph. We now reformulate this clearly, by defining the commonly made  "error in measurement height" at the beginning of Results section 3.1, and reformulating the sentence:

*"We conclude from this section, that considering T2m and T5m as fully equal temperatures is an invalid approximation: the difference between T2m and T5m is weak and within the measurement uncertainty on average over winter, but is not so during certain weather situations".*

Line 306-310: Can this argument be better generalised for instance by computing a diurnal cycle only for the anticyclonic conditions in the considered 4-year period? We will incorporate a statement on this aspect in the revised version of the manuscript, see for more detail the response to point 2.8.

Line 343: The discussion in the text "jumps" from Figure 4/5 to Figure 8, which is a bit confusing. Please readjusts the figure order or the text. This is now corrected in line with a complete reformulation of the Discussion.

Section 3.1.3 is difficult to follow. Could you please reformulate the Section to better distil the main message the authors would like to convey? For instance, reducing where possible references to "see above text" or if necessary, better explaining where a reader should focus, e.g. a figure or a particular subsection. This has been taken into account. Section 3.1.3 is now section 3.2.1 in the revised manuscript, and we invite the Referee to read this new section in the revised manuscript that we will provide shortly upon agreement of the Editor.

Line 369: "Secondly, we note that the analysed T5m is worse at Nivose stations". Worse than what? This is now better explained:

*"Secondly, we note that at Nivose stations, the analysed T5m performs poorly at night, and especially worse than the forecast at 5 m (at mid and high altitudes) and even then the background at 5 m (at high altitudes) (Figure 8 (b,d))."*

Line 383: "esp. ". please correct this. This has been corrected.

Line 387: "Mountainous areas are complex to instrument and model.", please clarify what "complex" means in this context. This sentence disappeared in the restructuration of the manuscript as redundant with the enhanced description of the research questions and their motivations in the Introduction.

Table 2: typos in "Biais", no closing parenthesis in figure caption. This is now corrected.

Line 495-497: Please rephrase. This has been rephrased.

*"It is therefore one of the least biased models according to the synthesis by Rudisill et al. (2024) and is close to the Canadian limited area model GEM-LAM evaluated by Vionnet et al. (2015), featuring a "0.5°C cold bias at high elevations" (Rudisill et al., 2024; Vionnet et al., 2015)."*

Line 513 – 520: This discussion is too much related to Meteo-France's models and using a lot of details that are not of interest to an external reader. The authors should give the information that would be of interest for the community, not the namelist settings used in operational AROME. This has been improved and we invite the referee to read the new version of the Discussion in the revised manuscript.

Line 588: "Correcting T2m biases would also enable the model's physical parameterisations to be improved.". Is it not the other way around, improvements in physical parameterisations reducing the T2m biases in the first guess and hence reducing the "activity" of the data assimilation system? What we meant by this sentence, is that a progress was needed to

more accurately characterize t2m, as a way to identify and henceforth limit "true" biases and error compensation. We rephrased for more clarity.

*"Having a more accurate T2m estimate, not affected by e.g. the error in measurement height, would enable a better knowledge of the true model biases, the formulation of relevant hypotheses for these biases and henceforth favor the improvement of the model's physical parameterisations. "*

Line 596: Is there any reference to previous work that could back up this hypothesis of misrepresented katabatic winds? Gouttevin et al. (2023) highlighted the possible role of Arome wind biases in the T2m bias of the model, but didn't specifically target thermal, especially katabatic winds. This is a quite recent hypothesis in our research group and has not been previously formulated in published literature.

Line 599-606: This long paragraph mixes perspectives and hypotheses for future work with conclusions from this work. I would suggest reorganising it so that the conclusions from this work are clearly divided from the perspectives. E.g. the recommendation to properly consider the height above the surface of the observations in assimilation and evaluation should come as a conclusion, whereas reducing temperature biases through improved model physics (using Isba-ES-DIFF etc.) should come as a perspective. We followed your suggestion and now better serapare the Conclusions (Section 5) that synthesize our main findings and recommendations on how to properly use mountain temperature measurements for assimilation and model evaluation, from the Perspectives that now belong to a dedicated subsection "4.1 Summary and general perspectives" opening the Discussion section. We refer the referee to the revised version of the manuscript for the Discussion, as reproducing this part here would be very long.